# Physics-Informed Bayesian Optimization of Variational Quantum Circuits

**Kim A. Nicoli**[1,2,3*]   **Christopher J. Anders**[3,4*]   **Lena Funcke**[1,2]   **Tobias Hartung**[5,6]
**Karl Jansen**[7]   **Stefan Kühn**[7]   **Klaus-Robert Müller**[3,4,8,9]   **Paolo Stornati**[10]
**Pan Kessel**[11]   **Shinichi Nakajima**[3,4,12]

[1]Transdisciplinary Research Area (TRA) Matter, University of Bonn, Germany
[2]Helmholtz Institute for Radiation and Nuclear Physics (HISKP)
[3]Berlin Institute for the Foundations of Learning and Data (BIFOLD)
[4] Technische Universität Berlin, Germany,    [5] Northeastern University London, UK
[6] Khoury College of Computer Sciences, Northeastern University, USA
[7] CQTA, Deutsches Elektronen-Synchrotron (DESY), Zeuthen, Germany
[8] Department of Artificial Intelligence, Korea University, Korea
[9]Max Planck Institut für Informatik, Saarbrücken, Germany
[10]Institute of Photonic Sciences, The Barcelona Institute of Science and Technology (ICFO)
[11]Prescient Design, gRED, Roche, Switzerland,    [12]RIKEN Center for AIP, Japan

## Abstract

In this paper, we propose a novel and powerful method to harness Bayesian optimization for Variational Quantum Eigensolvers (VQEs)—a hybrid quantum-classical protocol used to approximate the ground state of a quantum Hamiltonian. Specifically, we derive a *VQE-kernel* which incorporates important prior information about quantum circuits: the kernel feature map of the VQE-kernel exactly matches the known functional form of the VQE's objective function and thereby significantly reduces the posterior uncertainty. Moreover, we propose a novel acquisition function for Bayesian optimization called *Expected Maximum Improvement over Confident Regions* (EMICoRe) which can actively exploit the inductive bias of the VQE-kernel by treating regions with low predictive uncertainty as indirectly "observed". As a result, observations at as few as three points in the search domain are sufficient to determine the complete objective function along an entire one-dimensional subspace of the optimization landscape. Our numerical experiments demonstrate that our approach improves over state-of-the-art baselines.

## 1   Introduction

Quantum computing raises the exciting future prospect to efficiently tackle currently intractable problems in quantum chemistry [1], many-body physics [2], combinatorial optimization [3], machine learning [4], and beyond. Rapid progress over the last years has led to the development of the first noisy intermediate-scale quantum (NISQ) devices [5]; quantum hardware with several hundreds of qubits that can be harnessed to outperform classical computers on specific tasks (see, e.g., Ref. [6]). However, these tasks have been specifically designed to be hard on classical computers and easy on quantum computers, while being of no practical use. As we have entered the NISQ era, one of the grand challenges of quantum computing lies in the development of algorithms for NISQ devices that may exhibit a quantum advantage on a task of practical relevance.

---

*These authors contributed equally to the work.
Correspondence to `knicoli@uni-bonn.de` and `{anders,nakajima}@tu-berlin.de`

37th Conference on Neural Information Processing Systems (NeurIPS 2023).

One promising approach toward using NISQ devices is hybrid quantum-classical algorithms, such as VQEs [7, 8], which can be used to compute ground states of quantum Hamiltonians. VQEs can be seen as the quantum counterpart of neural networks: while classical neural networks model functions by parametric layers, the parametric quantum circuits used in VQEs represent variational wave functions by parametric quantum gates acting on qubits. During training, we aim to find a suitable choice for the variational parameters of the wave function such that the quantum mechanical expectation value of the Hamiltonian is minimized. From the optimization perspective, the quantum mechanical nature of the energy measurement is of little relevance. The training of VQEs can thus be regarded as a specific, albeit particularly challenging, noisy black-box optimization problem. Namely, we solve

$$\min_{\boldsymbol{x} \in [0, 2\pi)^D} f^*(\boldsymbol{x}), \tag{1}$$

where $f^*(\boldsymbol{x})$ is estimated from costly noisy observation, $y = f^*(\boldsymbol{x}) + \varepsilon$, on the quantum device.

Efficiently exploring the energy landscape of the VQE is crucial for successful optimization. This requires leveraging strong prior knowledge about the VQE objective function $f^*(\cdot)$. For instance, gradient-based optimization methods commonly utilize the parameter shift rule, which takes advantage of the specific structure of variational quantum circuits to compute gradients efficiently [9, 10]. Another approach is the *Nakanishi-Fuji-Todo* (NFT) method [11], a version of sequential minimal optimization (SMO) [12], which solves sub-problems sequentially. Nakanishi et al. [11] derived the explicit functional form of the VQE objective, enabling coordinate-wise global optimization with only two observations per iteration.

Given the highly non-trivial nature of classical optimization in VQE, machine learning represents an appealing tool to leverage the informative underlying physics toward a more effective search for a global optimum. Bayesian Optimization (BO) [13, 14] has been recently applied to VQE [15]. These methods have the distinct advantage that they can take the inherently noisy nature of the NISQ circuits into account. Unfortunately, these methods are yet to be competitive, especially in the high dimensional regime, because of their poor scalability and overemphasized-exploration behavior [16].

To overcome this limitation, this paper introduces a novel and powerful method for BO of VQEs, capitalizing on VQE-specific properties as physics-informed prior knowledge. To this end, we propose a *VQE-kernel*, designed with feature vectors that precisely align with the basis functions of the VQE objective. This incorporation of a strong inductive bias maximizes the statistical efficiency of Gaussian process (GP) regression and guarantees that GP posterior samples reside within the VQE function space. To further harness this powerful inductive bias, we present a novel acquisition function named *Expected Maximum Improvement over Confident Regions* (EMICoRe). EMICoRe operates by initially predicting the posterior variance and treating the points with low posterior variances as "observed" points. Subsequently, these indirectly observed points, alongside the directly observed ones, form the Confident Regions (CoRe) on which safe optimization of the GP mean is conducted to determine the current optimum. EMICoRe evaluates the expected maximum improvement of the best points in CoRe before and after the candidate points are observed. By utilizing EMICoRe, our approach combines the strengths of NFT [11] and BO, complementing each other in a synergistic manner. BO enhances the efficiency of NFT by replacing sub-optimal deterministic choices of next observation points, while the NFT procedure significantly constrains the exploration space of BO, leading to remarkable improvements in scalability.

Our numerical experiments demonstrate the performance gains of using the VQE kernel, and significant improvement of our NFT-with-EMICoRe approach in VQE problems with different Hamiltonians, different numbers of qubits, etc. As an additional contribution, we prove that two known important properties of VQE, the parameter shift rule [9] and the sinusoidal function-form [11], are equivalent, implying that they are not two different properties but two expressions of a single property.

**Related Work**   Numerous optimization methods for the VQE protocol have been proposed: gradient-based methods often rely on the parameter shift rule [9, 10], while NFT harnesses the specific functional form of the VQE objective [11] to establish SMO. BO has also been applied for VQE minimization [15]. Therein, it was shown that combining periodic kernel [17] and noisy expected improvement acquisition function [18] improves the performance of BO over the standard RBF kernel, and can make BO comparable to the state-of-the-art methods in the regime of small qubits and high observation noise.

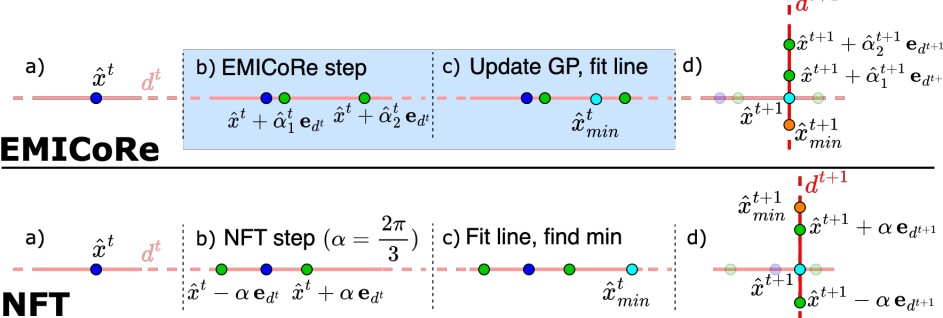

Figure 1: Illustration of EMICoRe (ours) and NFT [11] (baseline) procedures. In each step of NFT (bottom row), a) given the current best point $\hat{\boldsymbol{x}}^t$ and the direction $d^t$ to be explored, b) next observation points are chosen in a deterministic fashion. Then, c) the minimum along the line is found by the sinusoidal function fitting to the three points, and d) the next optimization step for the new direction $d^{t+1}$ starts from the found minimum $\hat{\boldsymbol{x}}^{t+1} = \hat{\boldsymbol{x}}^t_{\min}$. The EMICoRe procedure (top row) uses GP regression and BO in the steps highlighted by the light blue box: b) the next observation points are chosen by BO with the EMICoRe acquisition function based on the GP trained on the previous observations, and c) minimizing the predictive mean function of the updated GP with the new observations gives the best point $\hat{\boldsymbol{x}}^t_{\min}$.

BO is a versatile tool for black-box optimization in many applications [19, 20, 21, 22] including hyperparameter tuning of deep neural networks [23]. Most work on BO uses GP regression, and computes an acquisition function whose maximizer is suggested as the next observation point. Many acquisition functions, including lower confidence bound [24, 25], probability of improvement [26], Expected Improvement (EI) [27], entropy search [28, 29], and knowledge gradient [30] have been proposed. The most common acquisition function is EI, of which many generalizations exist: noisy EI (NEI) [18, 31] for dealing with observation noise, parallel EI [31] for batch sample suggestions, and EI per cost [23] for cost sensitivity. Our EMICoRe acquisition function is a generalization of noisy EI and parallel EI with the key novelty of introducing CoRe, which defines the indirectly observed points. Note the difference between the trust region [16] and CoRe: based on the predictive uncertainty, the former restricts the region to be explored, while the latter expands the observed points.

## 2 Preliminaries

### 2.1 Bayesian Optimization

Let $f^*(\cdot) : \mathcal{X} \mapsto \mathbb{R}$ be an unknown (black-box) objective function to be minimized. BO [13, 14] approximates the objective with a surrogate function $f(\boldsymbol{x}) \approx f^*(\boldsymbol{x})$, and suggests promising points to be observed in each step, such that the objective is likely to be improved considerably. A commonly used surrogate is the Gaussian process (GP) regression model [32] with one-dimensional Gaussian likelihood and GP prior,

$$p(y|\boldsymbol{x}, f(\cdot)) = \mathcal{N}_1(y; f(\boldsymbol{x}), \sigma^2), \qquad p(f(\cdot)) = \mathrm{GP}(f(\cdot); \nu(\cdot), k(\cdot, \cdot)), \qquad (2)$$

which is trained on the (possibly) noisy observations $y = f^*(\boldsymbol{x}) + \varepsilon$ made in the previous iterations. Here, $\sigma^2$ is the variance of observation noise $\varepsilon$, and $\nu(\cdot)$ and $k(\cdot, \cdot)$ are the prior mean and the kernel (covariance) functions, respectively. Throughout the paper, we set the prior mean function to be zero, i.e., $\nu(\boldsymbol{x}) = 0, \forall \boldsymbol{x} \in \mathcal{X}$. Let $\{\boldsymbol{X}, \boldsymbol{y}\}$ be $N$ training samples, where $\boldsymbol{X} = (\boldsymbol{x}_1, \ldots, \boldsymbol{x}_N) \in \mathcal{X}^N$ and $\boldsymbol{y} = (y_1, \ldots, y_N)^\top \in \mathbb{R}^N$. Since the GP prior is conjugate for the Gaussian likelihood[II], the posterior of the GP regression model (2) is also a GP, i.e., $p(f(\cdot)|\boldsymbol{X}, \boldsymbol{y}) = \mathrm{GP}(f(\cdot); \mu_{\boldsymbol{X}}(\cdot), s_{\boldsymbol{X}}(\cdot, \cdot))$, and thus, for arbitrary $M$ test inputs $\boldsymbol{X}' = (\boldsymbol{x}'_1, \ldots, \boldsymbol{x}'_M) \in \mathcal{X}^M$, the posterior of the function values $\boldsymbol{f}' = (f(\boldsymbol{x}'_1), \ldots, f(\boldsymbol{x}'_M))^\top \in \mathbb{R}^M$ is the $M$-dimensional Gaussian with its mean and covariance

---

[II]See Chap. 2 of Ref. [33] for the conjugacy in Bayesian inference.

analytically given as

$$p(\boldsymbol{f}'|\boldsymbol{X},\boldsymbol{y}) = \mathcal{N}_M(\boldsymbol{f}'; \boldsymbol{\mu}'_{\boldsymbol{X}}, \boldsymbol{S}'_{\boldsymbol{X}}), \quad \text{where} \tag{3}$$

$$\boldsymbol{\mu}'_{\boldsymbol{X}} = \boldsymbol{K}'^{\top}\left(\boldsymbol{K} + \sigma^2 \boldsymbol{I}_N\right)^{-1}\boldsymbol{y}, \qquad \boldsymbol{S}'_{\boldsymbol{X}} = \boldsymbol{K}'' - \boldsymbol{K}'^{\top}\left(\boldsymbol{K} + \sigma^2 \boldsymbol{I}_N\right)^{-1}\boldsymbol{K}'. \tag{4}$$

Here, $\boldsymbol{K} = k(\boldsymbol{X}, \boldsymbol{X}) \in \mathbb{R}^{N \times N}$, $\boldsymbol{K}' = k(\boldsymbol{X}, \boldsymbol{X}') \in \mathbb{R}^{N \times M}$, and $\boldsymbol{K}'' = k(\boldsymbol{X}', \boldsymbol{X}') \in \mathbb{R}^{M \times M}$ are the train, train-test, and test kernel matrices, respectively, where $k(\boldsymbol{X}, \boldsymbol{X}')$ denotes the kernel matrix evaluated at each column of $\boldsymbol{X}$ and $\boldsymbol{X}'$ such that $(k(\boldsymbol{X}, \boldsymbol{X}'))_{n,m} = k(\boldsymbol{x}_n, \boldsymbol{x}_m)$. Moreover, $\boldsymbol{I}_N \in \mathbb{R}^{N \times N}$ denotes the identity matrix, and the subscript $\boldsymbol{X}$ of posterior means and covariances specifies the input points on which the GP was trained (see Appendix B for details of GP regression).[III]

In the general BO framework [14], $M \geq 1$ input points to be observed next are chosen by (approximately) solving the following maximization problem in each iteration: $\max_{\boldsymbol{X}'} a_{\boldsymbol{X}^{t-1}}(\boldsymbol{X}')$, where $\boldsymbol{X}^t$ denotes the training input points observed until the $t$-th iteration, and $a_{\boldsymbol{X}}(\cdot)$ is an *acquisition function* computed based on the GP trained on the observations $\boldsymbol{y}$ at $\boldsymbol{X}$. The acquisition function evaluates the *promising-ness* of the new input points $\boldsymbol{X}'$, and should therefore give a high value if observing $\boldsymbol{X}'$ likely improves the current best score considerably. Common choices for the acquisition function are EI [27, 34], $a_{\boldsymbol{X}}^{\text{EI}}(\boldsymbol{x}') = \left\langle \max(0, \underline{f} - f') \right\rangle_{p(f'|\boldsymbol{X},\boldsymbol{y})}$, and its variants. Here, $\underline{f}$ denotes the current best observation, i.e., $\underline{f} = \min_{n \in \{1,\ldots,N\}} f(\boldsymbol{x}_n)$, and $\langle \cdot \rangle_p$ denotes the expectation value with respect to the distribution $p$. EI covers the case where the observation noise is negligible and only one sample is chosen in each iteration, i.e., $\sigma^2 \ll 1, M = 1$, and its analytic solution makes EI handy (see Appendix B). In the general case where $\sigma^2 > 0, M \geq 1$, generalizations of EI should be considered, such as NEI [18, 31],

$$a_{\boldsymbol{X}}^{\text{NEI}}(\boldsymbol{X}') = \left\langle \max(0, \min(\boldsymbol{f}) - \min(\boldsymbol{f}')) \right\rangle_{p(\boldsymbol{f}, \boldsymbol{f}'|\boldsymbol{X},\boldsymbol{y})}, \tag{5}$$

which appropriately takes into account the correlation between observations. NEI is estimated by quasi Monte Carlo sampling, and its maximization is approximately performed by sequentially adding a point to $\boldsymbol{X}'$ until $M$ points are collected.

## 2.2 Variational Quantum Eigensolver (VQE)

The VQE [7, 8] is a hybrid quantum-classical computing protocol for estimating the ground-state energy of a given quantum Hamiltonian for a $Q$-qubit system. The quantum computer is used to prepare a parametric quantum state $|\psi_{\boldsymbol{x}}\rangle$, which depends on $D$ angular parameters $\boldsymbol{x} \in \mathcal{X} = [0, 2\pi)^D$. This trial state $|\psi_{\boldsymbol{x}}\rangle$ is generated by applying $D'(\geq D)$ *quantum gate operations*, $G(\boldsymbol{x}) = G_{D'} \circ \cdots \circ G_1$, to an initial quantum state $|\psi_0\rangle$, i.e., $|\psi_{\boldsymbol{x}}\rangle = G(\boldsymbol{x})|\psi_0\rangle$. All gates $\{G_{d'}\}_{d'=1}^{D'}$ are unitary, and we assume in this paper that $x_d$ parametrizes only a single gate $G_{d'(d)}(x_d)$, where $d'(d)$ specifies the gate parametrized by $x_d$. Thus, $D$ of the $D'$ gates are parametrized by each entry of $\boldsymbol{x}$ *exclusively*.[IV] We consider parametric gates of the form $G_{d'}(x) = U_{d'}(x) = \exp(-ixP_{d'}/2)$, where $P_{d'}$ is an arbitrary sequence of the Pauli operators $\{\sigma_q^X, \sigma_q^Y, \sigma_q^Z\}_{q=1}^Q$ acting on each qubit at most once. This form covers not only the single-qubit gates such as $R_X(x) = \exp(-i\theta\sigma_q^X)$, but also the entangling gates, such as $R_{XX}(x) = \exp(-ix\sigma_{q_1}^X \circ \sigma_{q_2}^X)$ and $R_{ZZ}(x) = \exp(-ix\sigma_{q_1}^Z \circ \sigma_{q_2}^Z)$ for $q_1 \neq q_2$, which are commonly realized in trapped-ion quantum hardwares [35, 36].

The quantum computer evaluates the energy of the resulting quantum state $|\psi_{\boldsymbol{x}}\rangle$ by observing

$$y = f^*(\boldsymbol{x}) + \varepsilon, \qquad \text{where} \qquad f^*(\boldsymbol{x}) = \langle\psi_{\boldsymbol{x}}|H|\psi_{\boldsymbol{x}}\rangle = \langle\psi_0|G(\boldsymbol{x})^{\dagger}HG(\boldsymbol{x})|\psi_0\rangle \tag{6}$$

and $\dagger$ denotes the Hermitian conjugate. The observation noise $\varepsilon$ in our numerical experiments will only incorporate *shot noise*, and we will not consider the hardware-dependent errors induced by imperfect qubits, gates, and measurements. For each observation, multiple readout shots $N_{\text{shots}}$ are acquired to suppress the variance $\sigma^{*2}(N_{\text{shots}})$ of the noise $\varepsilon$. Since the observation $y$ is the sum of many random variables, it approximately follows a Gaussian distribution, according to the central limit theorem. The Gaussian likelihood in the GP regression model (2) therefore approximates the observation $y$ well if $f(\boldsymbol{x}) \approx f^*(\boldsymbol{x})$ and $\sigma^2 \approx \sigma^{*2}(N_{\text{shots}})$. With the quantum computer that

---

[III]Note that the subscript does not necessarily specify all dependencies. For example, $\boldsymbol{\mu}'_{\boldsymbol{X}}$ also depends on $\boldsymbol{y}$.

[IV]In Appendix E, we discuss how the theory and our method can be extended to the non-exclusive parametrization case.

provides noisy estimates of $f^*(\boldsymbol{x})$, a classical computer solves the minimization problem (1) and finds the minimizer $\hat{\boldsymbol{x}}$.

Several approaches, including stochastic gradient descent (SGD) [9, 10], SMO [11], and BO [15], have been proposed for solving the VQE optimization problem (1). Among others, state-of-the-art methods effectively incorporate unique properties of VQE. Let $\{\boldsymbol{e}_d\}_{d=1}^D$ be the standard basis.

**Proposition 1.** *[9] (Parameter shift rule) The VQE objective function $f^*(\cdot)$ in Eq. (6) for any parametric quantum circuit $G(\cdot)$, Hermitian operator $H$, and initial state $|\psi_0\rangle$ satisfies*

$$2\frac{\partial}{\partial x_d}f^*(\boldsymbol{x}) = f^*\left(\boldsymbol{x} + \frac{\pi}{2}\boldsymbol{e}_d\right) - f^*\left(\boldsymbol{x} - \frac{\pi}{2}\boldsymbol{e}_d\right), \quad \forall \boldsymbol{x} \in [0, 2\pi)^D, d = 1, \ldots, D. \tag{7}$$

Most SGD approaches rely on Proposition 1, which allows accurate gradient estimation from $2 \cdot D$ observations. Another useful property is used for tailoring SMO [12] to VQE:

**Proposition 2.** *[11] For the VQE objective function $f^*(\cdot)$ in Eq. (6) with any $G(\cdot)$, $H$, and $|\psi_0\rangle$,*

$$\exists \boldsymbol{b} \in \mathbb{R}^{3^D} \quad \text{such that} \quad f^*(\boldsymbol{x}) = \boldsymbol{b}^\top \cdot \mathbf{vec}\left(\otimes_{d=1}^D (1, \cos x_d, \sin x_d)^\top\right), \quad \forall \boldsymbol{x} \in [0, 2\pi)^D, \tag{8}$$

*where $\otimes$ and $\mathbf{vec}(\cdot)$ denote the tensor product and the vectorization operator for a tensor, respectively.*

Proposition 2 provides a strong prior knowledge on the VQE objective function $f^*(\cdot)$ — if we fix the function values at three general points along a coordinate axis, e.g., $\{\boldsymbol{x}, \boldsymbol{x} \pm \frac{2\pi}{3}\boldsymbol{e}_d\}$, for any $\boldsymbol{x}$ and $d$, then the whole function along the axis, i.e., $f^*(\boldsymbol{x} + \alpha \boldsymbol{e}_d), \forall \alpha \in [0, 2\pi)$ is fixed because it is a first-order sinusoidal function. Leveraging this property, a version of SMO was proposed [11]. This optimization strategy, named after its authors *Nakanishi-Fuji-Todo* (NFT), finds the global optimum in a single direction by sinusoidal fitting from three function values (two observed and one estimated) at each iteration, and showed state-of-the-art performance (see Appendix F.1 for details of NFT).

## 3 Proposed Method

In this section, we introduce our approach that combines BO and NFT by using a novel kernel and a novel acquisition function. After introducing these two ingredients, we propose our approach for VQE optimization. Lastly, we prove the equivalence between Propositions 1 and 2. We refer the reader to Appendix F, in particular Algorithms 1 to 3, for the pseudo-codes and further algorithmic details complementing the brief introduction to the algorithms presented in the following sections.

### 3.1 VQE Kernel

We first propose the following VQE kernel, and use it for the GP regression model (2):

$$k^{\text{VQE}}(\boldsymbol{x}, \boldsymbol{x}') = \sigma_0^2 \prod_{d=1}^D \left(\frac{\gamma^2 + 2\cos(x_d - x_d')}{\gamma^2 + 2}\right), \tag{9}$$

where $\sigma_0^2, \gamma^2 > 0$ are kernel parameters. $\sigma_0^2$ corresponds to the prior variance, while $\gamma^2$ controls the smoothness of the kernel. For $\gamma^2 = 1$, the VQE kernel is the product of Dirichlet kernels [37], each of which is associated with each direction $d$. We can show that this kernel (9) exactly gives the finite-dimensional feature space specified by Proposition 2 (the proof is given in Appendix D):

**Theorem 1.** *The VQE kernel (9) is decomposed as*

$$k^{VQE}(\boldsymbol{x}, \boldsymbol{x}') = \boldsymbol{\phi}(\boldsymbol{x})^\top \boldsymbol{\phi}(\boldsymbol{x}'), \text{ where } \boldsymbol{\phi}(\boldsymbol{x}) = \frac{\sigma_0}{(\gamma^2 + 2)^{D/2}}\mathbf{vec}\left(\otimes_{d=1}^D (\gamma, \sqrt{2}\cos x_d, \sqrt{2}\sin x_d)^\top\right).$$

Let $\mathcal{F}^{\text{VQE}}$ be the set of all possible VQE objective functions specified by Proposition 2. Theorem 1 guarantees that the support of the GP prior in Eq. (2) matches $\mathcal{F}^{\text{VQE}}$, if we use the VQE kernel function (9) with a prior mean function such that $\nu(\cdot) \in \mathcal{F}^{\text{VQE}}$. Thus, the VQE kernel drastically limits the expressivity of GP *without model misspecification*, which enhances statistical efficiency. A more important consequence of Theorem 1 is that Proposition 2 holds for any sample from the GP posterior with the VQE kernel (9). This implies that if GP is certain about three general points along an axis, it must be certain about the whole 1-D subspace going through those points. Theorem 1 can be generalized to the non-exclusive case, where each entry of $\boldsymbol{x}$ may parametrize multiple gates simultaneously (see Appendix E).

## 3.2 EMICoRe: Expected Maximum Improvement over Confident Regions

In GP regression, the predictive covariance does not depend on the observations $\boldsymbol{y}$ (see Eq. (4)), implying that, for given training inputs $\boldsymbol{X} \in \mathcal{X}^N$, we can compute the predictive variance $s_{\boldsymbol{X}}(\boldsymbol{x}, \boldsymbol{x})$ at any $\boldsymbol{x} \in \mathcal{X}$ *before observing the function values*. Let us define *Confident Regions* (CoRe) as

$$\mathcal{Z}_{\boldsymbol{X}} = \left\{ \boldsymbol{x} \in \mathcal{X}; s_{\boldsymbol{X}}(\boldsymbol{x}, \boldsymbol{x}) \leq \kappa^2 \right\}, \tag{10}$$

which corresponds to the set on which the predicted uncertainty by GP is lower than a threshold $\kappa$. For sufficiently small $\kappa$, an appropriate kernel (which does not cause model misspecification), and a sufficiently weak prior, the GP predictions on CoRe are already accurate, and therefore CoRe can be regarded as "observed points." This leads to the following acquisition function, which we call Expected Maximum Improvement over Confident Regions (EMICoRe):

$$a^{\mathrm{EMICoRe}}(\boldsymbol{X}') = \tfrac{1}{M} \left\langle \max \left( 0, \min_{\boldsymbol{x} \in \mathcal{Z}_{\boldsymbol{X}}} f(\boldsymbol{x}) - \min_{\boldsymbol{x} \in \mathcal{Z}_{\widetilde{\boldsymbol{X}}}} f(\boldsymbol{x}) \right) \right\rangle_{p(f(\cdot)|\boldsymbol{X}, \boldsymbol{y})}, \tag{11}$$

where $\widetilde{\boldsymbol{X}} = (\boldsymbol{X}, \boldsymbol{X}') \in \mathcal{X}^{N+M}$ denotes the augmented training set with the new input points $\boldsymbol{X}' \in \mathcal{X}^M$. This acquisition function evaluates the expected maximum improvement (per new observation point) when CoRe is expanded from $\mathcal{Z}_{\boldsymbol{X}}$ to $\mathcal{Z}_{\widetilde{\boldsymbol{X}}}$ by observing the objective at the new input points $\boldsymbol{X}'$. EMICoRe can be seen as a generalization of existing methods. If CoRe consists of the training points, it reduces to NEI [18]. If we set $\kappa \to \infty$ so that the whole space is in the CoRe, and the random function $f(\cdot)$ in Eq. (11) is replaced with its predictive mean of the current (first term) and the updated (second term) GP, EMICoRe reduces to knowledge gradient (KG) [30]. Thus, KG can be seen as a version of EMICoRe that ignores the uncertainty of the updated GP.

### 3.2.1 NFT-with-EMICoRe

We enhance the state-of-the-art NFT approach [11] with the VQE kernel and EMICoRe. We start from a brief overview of the NFT algorithm (detailed algorithms of NFT, NFT-with-EMICoRe, and the EMICoRe subroutine are given in Appendix F).

**NFT:** First, we initialize with a random point $\hat{\boldsymbol{x}}^0$ with $\hat{y}^0 = f^*(\hat{\boldsymbol{x}}^0) + \varepsilon$. Then, for each iteration step $t$, we proceed as follows:

1. Select an axis $d \in \{1, \ldots, D\}$ sequentially or randomly and observe the objective $\boldsymbol{y}' = (y'_1, y'_2)^\top$ at *deterministically* chosen two points $\boldsymbol{X}' = (\boldsymbol{x}'_1, \boldsymbol{x}'_2) = \{\hat{\boldsymbol{x}}^{t-1} - 2\pi/3\boldsymbol{e}_d, \hat{\boldsymbol{x}}^{t-1} + 2\pi/3\boldsymbol{e}_d\}$ along the axis $d$.
2. Fit the sinusoidal function $\widetilde{f}(\theta) = c_0 + c_1 \cos \theta + c_2 \sin \theta$ to the two new observations $y'_1, y'_2$ as well as the previous best estimated score $\hat{y}^{t-1}$. The optimal shift $\hat{\theta}$ that minimizes $\widetilde{f}(\theta)$ is analytically computed, which is used to get the new optimum $\hat{\boldsymbol{x}}^t = \hat{\boldsymbol{x}}^{t-1} + \hat{\theta}\boldsymbol{e}_d$.
3. The best score is updated as $\hat{y}^t = \widetilde{f}(\hat{\boldsymbol{x}}^t)$.

We stress that if the observation noise is negligible, i.e., $y \approx f(\boldsymbol{x})$, each step of NFT reaches the global optimum in the one-dimensional subspace along the chosen axis $d$, and thus performs SMO, see Proposition 2. In this case, the choice of the two new observation points $\boldsymbol{X}'$ is not important, as long as any pair of the three points are not exactly the same. However, when the observation noise is significant, the estimated global optimum in the chosen subspace is not necessary accurate, and the accuracy highly depends on the choice of the new observation points $\boldsymbol{X}'$. In addition, errors can be accumulated in the best score $\hat{y}^t$, and therefore an additional measurement needs to be performed at $\hat{\boldsymbol{x}}^t$ after a certain iteration interval.

**NFT-with-EMICoRe (ours):** We propose to apply BO to NFT by using the VQE kernel and EMICoRe. Specifically, we use the GP regression model with the VQE kernel as a surrogate for BO, and choose new observation points $\boldsymbol{X}'$ by using the EMICoRe acquisition function. NFT-EMICoRe starts from $T_{\mathrm{NFT}}$ NFT iterations until GP gets informative with a sufficient number of observations. After this initial phase, we proceed for each iteration $t$ as follows:

1. Select an axis $d \in \{1, \ldots, D\}$ sequentially, and new observation points $\boldsymbol{X}'$ by BO with EMICoRe, based on the previous optimum $\hat{\boldsymbol{x}}^{t-1}$, the previous training data $\{\boldsymbol{X}^{t-1}, \boldsymbol{y}^{t-1}\}$, and the current CoRe threshold $\kappa^t$ (this subroutine for EMICoRe will be explained below).

2. We observe $\boldsymbol{y}'$ at the new points $\boldsymbol{X}'$ chosen by EMICoRe, and train GP with the updated training data $\boldsymbol{X}^t = (\boldsymbol{X}^{t-1}, \boldsymbol{X}'), \boldsymbol{y}^t = (\boldsymbol{y}^{t-1\top}, \boldsymbol{y}'^\top)^\top$.

3. The subspace optimization is performed by fitting a sinusoidal function $\widetilde{f}(\theta)$ to the GP posterior means $\boldsymbol{\mu} = (\mu(\hat{\boldsymbol{x}}^{t-1} - 2\pi/3\boldsymbol{e}_d), \mu(\hat{\boldsymbol{x}}^{t-1}), \mu(\hat{\boldsymbol{x}}^{t-1} + 2\pi/3\boldsymbol{e}_d))^\top$ at three points. With the analytic minimum of the sinusoidal function, $\hat{\theta} = \arg\min_\theta \widetilde{f}(\theta)$, the current optimum is computed: $\hat{\boldsymbol{x}}^t = \hat{\boldsymbol{x}}^{t-1} + \hat{\theta}\boldsymbol{e}_d$.

4. For the current best score, we simply use the GP posterior mean $\hat{\mu}^t = \mu(\hat{\boldsymbol{x}}^t)$ and we set the new CoRe threshold to

$$\kappa^{t+1} = \frac{\hat{\mu}^{t-T_{\text{Ave}}} - \hat{\mu}^t}{T_{\text{Ave}}}. \tag{12}$$

Note that the GP posterior mean function lies in the VQE function space $\mathcal{F}^{\text{VQE}}$, and therefore the fitting is done without error, and the choice of the three points in Step 3 does not affect the accuracy. Note also that the CoRe threshold $\kappa^{t+1}$ adjusts the required accuracy to the average reduction of the best score over the $T_{\text{Ave}}$ latest iterations—in the early phase where the energy $\hat{\mu}^t$ decreases steeply, a large $\kappa$ encourages crude optimization, while in the converging phase where the energy decreases slowly, a small $\kappa$ enforces accurate optimization. Figure 1 illustrates the procedures of NFT and our EMICoRe approach.

The EMICoRe subroutine receives the previous optimum $\hat{\boldsymbol{x}}^{t-1}$, the previous training data $\{\boldsymbol{X}^{t-1}, \boldsymbol{y}^{t-1}\}$, and the current CoRe threshold $\kappa^t$, and returns the new points $\boldsymbol{X}'$ that maximizes EMICoRe. We fix the number of new samples to $M = 2$, and perform grid search along the chosen direction $d$. To this end,

1. We prepare $J_{\text{SG}}(J_{\text{SG}} - 1)$ combinations of $J_{\text{SG}}$ search grid points $\{\hat{\boldsymbol{x}}^{t-1} + \alpha_j \boldsymbol{e}_d\}_{j=1}^{J_{\text{SG}}}$, where $\boldsymbol{\alpha} = (\alpha_1, \dots \alpha_{J_{\text{SG}}})^\top = \frac{2\pi}{J_{\text{SG}}+1} \cdot (1, \dots, J_{\text{SG}})^\top$, as a candidate set $\mathcal{C} = \{\breve{\boldsymbol{X}}^j \in \mathbb{R}^{D \times 2}\}_{j=1}^{J_{\text{SG}}(J_{\text{SG}}-1)}$.

2. For each candidate $\breve{\boldsymbol{X}} \in \mathcal{C}$, we compute the *updated* GP posterior variance $s_{\widetilde{\boldsymbol{X}}}(\boldsymbol{x}, \boldsymbol{x}), \forall \boldsymbol{x} \in \boldsymbol{X}^{\text{Grid}}$, where $s_{\widetilde{\boldsymbol{X}}}(\cdot, \cdot)$ is the posterior covariance function of the GP trained on the augmented training points $\widetilde{\boldsymbol{X}} = (\boldsymbol{X}^{t-1}, \breve{\boldsymbol{X}})$, and $\boldsymbol{X}^{\text{Grid}} = \{\hat{\boldsymbol{x}}^{t-1} + \alpha_j \boldsymbol{e}_d\}_{j=1}^{J_{\text{OG}}}$ with $\boldsymbol{\alpha} = (\alpha_1, \dots \alpha_{J_{\text{OG}}})^\top = \frac{2\pi}{J_{\text{OG}}+1} \cdot (1, \dots, J_{\text{OG}})^\top$ are $J_{\text{OG}}$ grid points along the axis $d$.

3. We obtain a discrete approximation to the updated CoRe as $\mathcal{Z}_{\widetilde{\boldsymbol{X}}} = \{\boldsymbol{x} \in \boldsymbol{X}^{\text{Grid}}; s_{\widetilde{\boldsymbol{X}}}(\boldsymbol{x}, \boldsymbol{x}) \leq \kappa^2\}$. For simplicity, we approximate the previous CoRe to the previous optimum, i.e., $\mathcal{Z}_{\boldsymbol{X}^{t-1}} = \{\hat{\boldsymbol{x}}^{t-1}\}$. After computing the mean and the covariance of the previous GP posterior, $p(\hat{f}, \boldsymbol{f}^{\text{test}} | \boldsymbol{X}^{t-1}, \boldsymbol{y}^{t-1})$, at the previous best point $\hat{\boldsymbol{x}}^{t-1}$ and the updated CoRe points (as the test set $\boldsymbol{X}^{\text{test}} = \mathcal{Z}_{\widetilde{\boldsymbol{X}}}$)—which is $\widetilde{D}$-dimensional Gaussian for $\widetilde{D} = |\mathcal{Z}_{\boldsymbol{X}^{t-1}} \cup \mathcal{Z}_{\widetilde{\boldsymbol{X}}}| = 1 + |\mathcal{Z}_{\widetilde{\boldsymbol{X}}}|$—we estimate $a_{\boldsymbol{X}^{t-1}}^{\text{EMICoRe}} = \frac{1}{M} \langle \max\{0, \hat{f} - \min(\boldsymbol{f}^{\text{test}})\} \rangle_{p(\hat{f}, \boldsymbol{f}^{\text{test}} | \boldsymbol{X}^{t-1}, \boldsymbol{y}^{t-1})}$ by quasi Monte Carlo sampling.

The subroutine iterates this process for all candidates, and returns the best one,

$$\boldsymbol{X}' = \underset{\breve{\boldsymbol{X}} \in \mathcal{C}}{\arg\max}\, a_{\boldsymbol{X}^{t-1}}^{\text{EMICoRe}}(\breve{\boldsymbol{X}}).$$

**Parameter Setting** Our approach has the crucial advantage that the sensitive parameters can be automatically tuned. The kernel smoothness parameter $\gamma$ is optimized by maximizing the marginal likelihood of the GP, and the CoRe threshold $\kappa$ is set to the average energy decrease of the last iterations as explained above.[V] The noise variance $\sigma^2$ is set by observing $f^*(\boldsymbol{x})$ several times at several random points and estimating $\sigma^2 = \hat{\sigma}^{*2}(N_{\text{shots}})$. For the GP prior, the zero mean function $\nu(\boldsymbol{x}) = 0, \forall \boldsymbol{x}$ is used, and the prior variance $\sigma_0^2$ is roughly set so that the absolute value of the ground-state energy is in the same order as $\sigma_0$. Other parameters, including the number of grid points for search and CoRe discretization, should be set to sufficiently large values. See Appendix F for more details of parameter setting.

---

[V] In Appendix F.4, we investigate other heuristics for the $\kappa$ update, and find that the default update rule (12) performs comparably to the best heuristic.

### 3.3 Equivalence Between VQE Properties

Our method, NFT-with-EMICoRe, adapts BO to the VQE problem by harnessing one of the useful properties of VQE, namely the highly constrained objective function form (Proposition 2). One may now wonder if we can further improve NFT-EMICoRe by harnessing the other property, e.g., by regularizing GP so that its gradients follow the parameter shift rule (Proposition 1). We give a theory that answers to this question. Although the two properties were separately derived by analyzing the VQE objective (6), they are actually mathematically equivalent (the proof is given in Appendix G):

**Theorem 2.** *For any periodic function* $f^* : [0, 2\pi)^D \mapsto \mathbb{R}$, *Eq.* (7) $\Leftrightarrow$ *Eq.* (8).

This means that any sample from the GP prior with the VQE kernel (and any prior mean such that $\nu(\cdot) \in \mathcal{F}^{\mathrm{VQE}}$) already satisfies the parameter shift rule. Theorem 2 implies that the parameter shift rule and the VQE function form are not two different properties, but two expressions of a *single* property, which is an important result that—to our knowledge—was not known in the literature.

## 4 Experiments

We numerically demonstrate the performance of our approach for several setups, where the goal is to find the ground state of the quantum Heisenberg Hamiltonian

$$H = - \left[ \sum_{j=1}^{Q-1} (J_X \sigma_j^X \sigma_{j+1}^X + J_Y \sigma_j^Y \sigma_{j+1}^Y + J_Z \sigma_j^Z \sigma_{j+1}^Z) + \sum_{j=1}^{Q} (h_X \sigma_j^X + h_Y \sigma_j^Y + h_Z \sigma_j^Z) \right], \quad (13)$$

where $\{\sigma_j^X, \sigma_j^Y, \sigma_j^Z\}$ represent the Pauli matrices applied on the qubit in site $j$. The Heisenberg Hamiltonian, which represents a standard benchmarks of high practical relevance, is commonly used for evaluating the VQE performance (see, e.g., [38]),[VI] and its ground-truth ground state $|\psi_{\mathrm{GS}}\rangle$ along with the ground-state energy $\underline{f}^* = \langle \psi_{\mathrm{GS}}| H |\psi_{\mathrm{GS}}\rangle$—which gives a lower-bound of the VQE objective (6)—can be analytically computed for small $Q$. For the variational quantum circuit $G(\boldsymbol{x})$, we use the $L$-layered `Efficient SU(2)` circuit (see Appendix C) with the open boundary, for which the search domain of the angular variables is $\boldsymbol{x} \in [0, 2\pi)^D$ with $D = (2 + (L \cdot 2)) \cdot Q$. We classically simulate the quantum computation with the Qiskit [43] library, and our Python implementation along with detailed tutorials on how to reproduce the results is publicly available on GitHub [44] at *https://github.com/emicore/emicore*.

To measure the optimization performance, i.e., the quality of the final solution $\hat{\boldsymbol{x}}^T$ after $T$ iterations, we use two metrics: the *true* achieved lowest energy after $T$ steps, $\mathrm{ENG} \equiv f^*(\hat{\boldsymbol{x}}^T) = \langle \psi_0| G(\hat{\boldsymbol{x}}^T)^\dagger H G(\hat{\boldsymbol{x}}^T) |\psi_0\rangle$, which is evaluated by simulating the noiseless observation with $N_{\mathrm{shots}} \to \infty$, and the fidelity $\mathrm{FID} \equiv \langle \psi_{\mathrm{GS}}|\psi_{\hat{\boldsymbol{x}}^T}\rangle = \langle \psi_{\mathrm{GS}}| G(\hat{\boldsymbol{x}}^T) |\psi_0\rangle$, which measures how similar the best solution is to the true ground state. As the cost of observations, we count the total number of observed points, ignoring the cost of classical computation. We test each method 50 times, using the same set of initial points, for each method, for fair comparison. The initial points were randomly drawn from the uniform distribution in $[0, 2\pi)^D$, for fair comparisons. Details of the VQE circuits and the experimental settings can be found in Appendices C and H, respectively.

### 4.1 VQE-kernel Analysis

We first investigate the benefit of using the VQE-kernel for the following optimization problem: the task is to minimize the VQE objective (6) for the Ising Hamiltonian, a special case of the Heisenberg Hamiltonian with the coupling parameters set to $J_X = -1$, $J_Y = J_Z = 0, h_X = h_Y = 0, h_Z = -1$, with the $(L = 3)$-layered quantum circuit with $(Q = 3)$-qubits. Namely, we solve the problem (1) in the $(D = 2(L + 1)Q = 24)$-dimensional space. We use the standard BO procedure with GP regression, and compare our VQE-kernel with the Gaussian radial basis function (RBF) [32] and the periodic kernel [17], where the EI acquisition function is maximized by L-BFGS [45]. Figure 2 shows the achieved energy (left) and the fidelity (right) with different kernels. In each panel, the left

---

[VI]Spin chain Hamiltonians are widely studied in condensed matter physics [39], and generalized spin chains also emerge from the lattice discretization of field theories in low dimensions (see, e.g., Ref. [40] or Refs. [41, 42] for reviews).

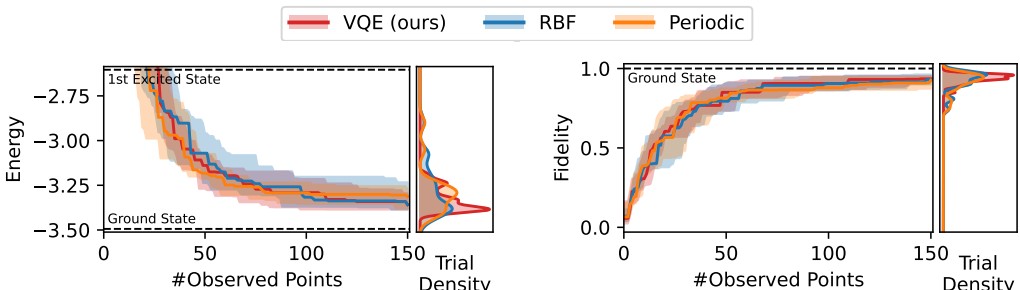

Figure 2: Comparison of our VQE-kernel (red) to the RBF and the periodic kernel benchmarks (blue and orange) in the VQE optimization using the standard BO procedure, for the Ising Hamiltonian with the ($L = 3$)-layered ($Q = 3$)-qubits quantum circuit. The search domain dimension is $D = 24$, and $N_{\text{shots}} = 1024$ readout shots are taken for each observation. The energy (left) and the fidelity (right) are plotted, and in each plot, optimization progress is shown with the median (solid) and the 25- and 75-th percentiles (shadows) over 50 trials. The portrait square shows the distribution of the final solution after 150 observations have been performed.

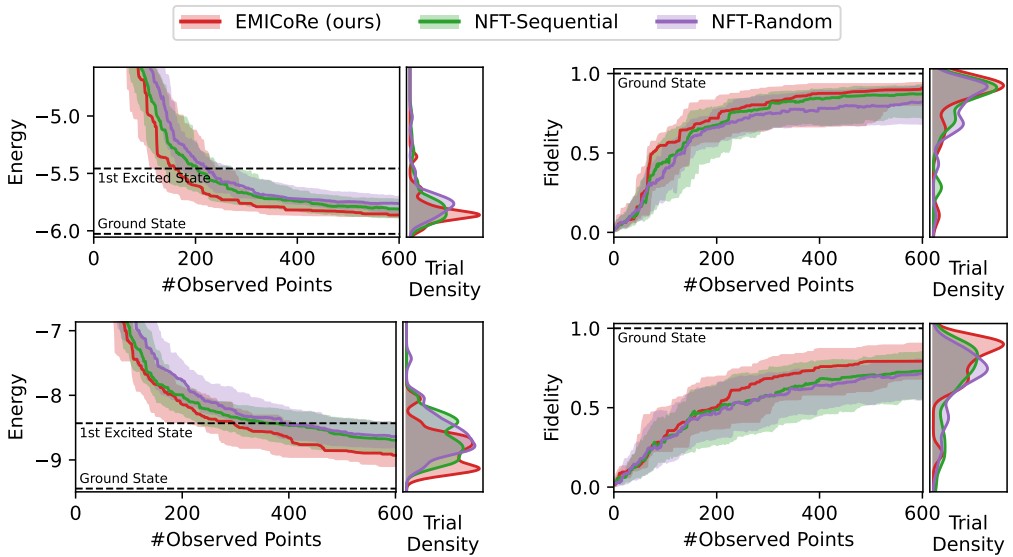

Figure 3: Comparison (in the same format as Figure 2) between our EMICoRe (red) and the NFT baselines (green and purple) in the VQE for the Ising (top row) and Heisenberg (bottom row) Hamiltonians with the ($L = 3$)-layered ($Q = 5$)-qubits quantum circuit (thus, $D = 40$) and $N_{\text{shots}} = 1024$. We confirmed for the Ising Hamiltonian that **longer optimization by EMICoRe up to 6000 observed points reaches the ground state with** $98\%$ **fidelity (see Appendix I.1)**.

plot shows the progress of optimization (median as a solid curve and the 25- and 75-th percentiles as shadows) as a function of the observation cost, i.e., the number of observed points after corresponding iterations. The portrait square on the right shows the distribution (by kernel density estimation [46]) of the best solutions after 150 observations have been performed. We see that the proposed VQE kernel (red), which achieves $0.93 \pm 0.05$ fidelity, converges more stably than the baseline RBF and periodic kernels, both of which achieve $0.90 \pm 0.09$ fidelity. Therefore, the VQE kernel leads to better statistical efficiency albeit its effect appears rather mild in the regime of a small number of qubits.

### 4.2 NFT-with-EMICoRe Analysis

We now evaluate our NFT-with-EMICoRe approach and show that this improves the state-of-the-art baselines of NFT [11]. Specifically, we compare our method with two versions of NFT: NFT-Sequential updates along each axis sequentially, while NFT-Random chooses the axis randomly. Figure 3 shows the results of VQE for the Ising Hamiltonian (top row) and the Heisenberg Hamiltonian (bottom row) with the parameters set to $J_X = J_Y = J_Z = 1$ and $h_X = h_Y = h_Z = 1$. For both

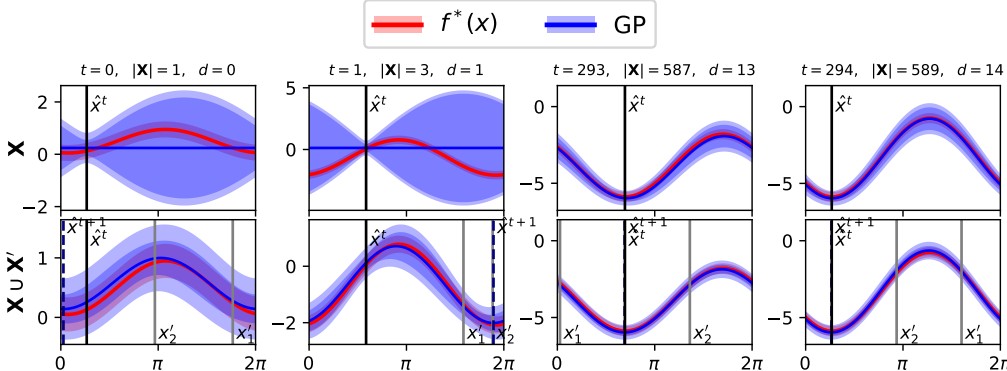

Figure 4: Evolution of the GP in NFT-with-EMICoRe. The posterior mean (blue solid) with uncertainty (blue shadow) along the direction $d$ to be optimized in the steps $t = 0, 1, 293, 294$ (columns) are shown before (top row) and after (bottom row) the chosen two new points $\boldsymbol{X}' = (\boldsymbol{x}'_1, \boldsymbol{x}'_2)$ are observed. The red solid curve shows the true energy $f^*(\boldsymbol{x})$. On the top, the number of observed samples $|\boldsymbol{X}|$ until step $t - 1$ and the direction $d$ are shown.

Hamiltonians, we use $(L = 3)$-layered $(Q = 5)$-qubits variational quantum circuits, thus $D = 40$, with each observation performed by $N_{\text{shots}} = 1024$ readout shots. The figure shows that our NFT-with-EMICoRe method consistently outperforms the baselines in terms of both achieved energy (left) and fidelity (right). Moreover, the variance over the trials is reduced, implying its robustness against the initialization. We conducted additional experiments to answer the important questions of whether our approach converges to the ground state with high fidelity, and how the Hamiltonian, the number of qubits, and the number of readout shots affect the optimization performance. The results reported in Appendix I show that our NFT-with-EMICoRe method **reaches the ground state with 98% fidelity after 6000 observed points**, see Appendix I.1, and consistently outperforms the baseline methods in various settings of Hamiltonians, qubits, and the number of readout shots, see Appendices I.2 and I.3. Thus, we conclude that our approach of combining NFT and BO with the VQE-kernel and the EMICoRe acquisition function is suited for efficient VQE optimization, dealing with different levels of observation noise.

Figure 4 shows the evolution of the GP during optimization by NFT-with-EMICoRe. The blue curve and the shadow show the posterior mean (solid) and the uncertainty (shadow) at four different steps $t$ (columns). The top and the bottom rows show the GP before and after the two new points $\boldsymbol{X}' = (\boldsymbol{x}'_1, \boldsymbol{x}'_2)$, chosen by EMICoRe, are observed. The red solid curve shows the true energy $f^*(\boldsymbol{x})$. We observe the following: In the early phase (the left two columns), GP is uncertain except at the current optimum $\hat{\boldsymbol{x}}^t$ before new points are observed; After observing the chosen two points $\boldsymbol{X}'$, GP is certain on the whole subspace, thanks to the VQE kernel; and in the converging phase (the right two columns), GP is certain over the whole subspace even before new observations, and the fine-tuning of the angular variable $\boldsymbol{x}$ is performed.

## 5 Conclusion

Efficient, noise-resilient algorithms for optimizing hybrid quantum-classical algorithms are pivotal for the NISQ era. In this paper, we propose EMICoRe, a new Bayesian optimization approach, in combination with a novel VQE-kernel which leverages strong inductive biases from physics. Our physics-informed search with EMICoRe achieves faster and more stable convergence to the minimum energy, thus identifying optimal quantum circuit parameters. Remarkably, the physical inductive bias makes the optimization more resilient to observation noise. The insight that more physical information helps to gain a better practical understanding of quantum systems is a starting point for further algorithmic improvements and further exploration of other possible physical biases. This may ultimately lead to more noise-resilient algorithms for quantum circuits, thus enabling the accessibility of larger-scale experiments in the quantum computing realm. In future work, we plan to investigate the effect of hardware-dependent noise, and test our approach on real quantum devices.

## Acknowledgments

The authors thank the referees for their insightful comments, which significantly improved the paper. This work was partially supported by the German Ministry for Education and Research (BMBF) as BIFOLD – Berlin Institute for the Foundations of Learning and Data (BIFOLD23B), the Einstein Research Unit (ERU) Quantum Project (ERU-2020-607), and the European Union's HORIZON MSCA Doctoral Networks programme and the AQTIVATE project (101072344). This work is supported with funds from the Ministry of Science, Research and Culture of the State of Brandenburg within the Centre for Quantum Technologies and Applications (CQTA). PS acknowledges support from ERC AdG NOQIA; MICIN/AEI (PGC2018-0910.13039/501100011033, CEX2019-000910-S/10.13039/501100011033, Plan National FIDEUA PID2019-106901GB-I00, FPI). K.-R.M. was partly supported by the Institute of Information & Communications Technology Planning & Evaluation (IITP) grants funded by the government (MSIT) (No. 2019-0-00079, Artificial Intelligence Graduate School Program, Korea University and No. 2022-0- 00984, Development of Artificial Intelligence Technology for Personalized Plug-and-Play Explanation and Verification of Explanation).

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

## Limitations

One significant limitation of the approach presented in this paper, particularly in the context of Variational Quantum Eigensolvers (VQEs), relates to the scalability of Gaussian Processes (GPs). When a large number of points is added to the GP training set through additional observations, the computational scalability becomes a challenge, especially in scenarios involving a large number of observations. However, we consider a potential solution to address this issue by imposing a fixed limit on the training sample size. This approach involves removing previously observed points and replacing them with newer ones. We hypothesize that by leveraging the information from the CoRe, the newly added points would contain significantly more valuable information, making previous observations less informative. Consequently, removing those points from the training set would mitigate the inherent scalability problem associated with GPs. Exploring this idea further is an avenue for future research. In addition, in the current version of our proposed NFT-with-EMICoRe, we limited the choice of new observation points to two points along a sequentially chosen axis, which is clearly sub-optimal. We will extend our approach for more flexible choices, e.g., by including sets of points along different directions and different numbers of points to the candidate set, in future work.

Another limitation is related to the current constraints of VQEs and Noisy Intermediate-Scale Quantum (NISQ) devices. The execution of quantum computations on NISQ devices is currently restricted, in particular by the coherence time of the qubits and the number of operations required to execute the algorithm. Consequently, the measurements on a quantum computer are susceptible to errors, which is recognized as one of the most challenging obstacles in the field. Although error mitigation techniques have been proposed [47], developing hybrid classical-quantum algorithms that are more resilient to the inherent noise of quantum computers remains an open area of research.

## Broader Impact

This work is a remarkably successful example of the synergistic combination of theories and techniques developed in physics and machine learning communities. Namely, the strong physical prior knowledge of the VQE objective function is well incorporated in the kernel machine, leading to our novel VQE kernel, while a specialized sequential minimal optimization—the NFT algorithm—developed in the physics community is adapted to the Bayesian Optimization (BO) framework, mitigating the suboptimality of NFT and the scalability issue (in terms of the search domain dimensionality) of BO in a complementary manner. Our novel EMICoRe acquisition function plays an important role: it exploits the correlations between the function values on points, which are not necessarily in the neighborhood of each other, by leveraging a new concept of confident regions. This technique can be applied to general optimization problems where the prior knowledge implies such non-trivial correlations. In addition, the equivalence of the parameter shift rule and the VQE function form, which we have proved without using any physics-specific knowledge, can facilitate theoretical developments both in physics and machine learning communities.

Regarding the societal impact, the authors have thoroughly considered the potential negative consequences of the proposed work and have found none.

## A  Extended Related Work

Since the VQE protocol was first introduced [7], many optimization algorithms have been proposed for minimizing the VQE objective. For gradient-based optimization, the parameter shift rule allows for an efficient gradient computation on NISQ devices [9, 10]. Making use of the analytic form of the gradient in typical parametric ansatz circuits, this approach avoids estimating the gradient using finite differences, which would be challenging for NISQ hardware due to the limited accuracy that can be achieved due to noise.

The Nakanishi-Fuji-Todo (NFT) method [11] harnesses the specific function-form of the VQE objective to establish Sequential Minimal Optimization (SMO) and showed the state-of-the-art performance. The authors focused on the case where the parametric gates are of the form $U(x_d) = \exp(-ix_d P/2)$ with angular parameters $x \in \mathbb{R}^D$ and operators $P$ fulfilling $P^2 = I$ and derived an explicit function-form of the VQE objective. The resulting function form implies that, by keeping all parameters (angles) in the circuit fixed except for a single one, one can identify the complete

objective function in the corresponding one-dimensional subspace by observing only three points, and the global minimum in the subspace can be analytically found. NFT uses this property and performs SMO by choosing a one-dimensional subspace sequentially or randomly until convergence, providing an efficient and stable algorithm suited for NISQ devices.

BO is a versatile tool for black-box optimization with its applications including engineering system design [19], drug design [20], material design [21], and reinforcement learning [22]. Recently, it started to be extensively used for hyperparameter tuning of deep neural networks [23]. Most work on BO uses the GP regression model, computes an acquisition function, which evaluates the *promising-ness* of the next candidate points, and suggests its maximizer as the set of next observation points. Many acquisition functions have been proposed. Lower (upper for maximization problems) confidence bound [24, 25] optimistically gives high acquisition values at the points with low predictive mean and high uncertainty. Probability of improvement [26] and expected improvement (EI) [27, 34] evaluate the probability and the expectation value that the next point can improve the previous optimum. Entropy search [28, 29] searches the point where the entropy at the minimizer is expected to be minimized. Knowledge gradient [30] allows final uncertainty and estimates the improvement of the optimum of the predictions before and after the next sample is included in the training data. The most common acquisition function is EI, and many generalizations have been proposed. Noisy EI (NEI) [18] considers the observation noise and takes the correlations between observations into account, parallel EI [31] considers the case where a batch of new samples are to be suggested, and EI per cost (or per second if only the computation time matters) [23] penalizes the acquisition function value based on the (estimated) observation cost.

BO has also been applied to VQE minimization [15]. It was shown that combining the periodic kernel [17] and NEI acquisition function [18] significantly improves the performance of BO with the plain RBF kernel and EI, thus making BO comparable to the state-of-the-art methods in the regime of small qubits and high observation noise. Our approach with Expected Maximum Improvement over Confident Regions (EMICoRe) has similarities to existing methods and can be seen as a generalization of them. The key novelty is the introduction of Core Regions (CoRe), which defines the indirectly observed points. Note the difference between the trust region [16] and CoRe: Based on the predictive uncertainty, the former restricts the regions to be explored, while the latter expands the observed points.

For completeness, we note that other machine learning techniques from reinforcement learning [48] and deep generative models [49] have also been applied to improve the classical optimization schemes of VQEs.

## B  Details of Gaussian Process (GP) Regression and Bayesian Optimization (BO)

In the following, we introduce GP, GP regression, and BO with an uncompressed notation.

### B.1  Gaussian Process Regression

A GP [32] is an infinite-dimensional generalization of multivariate Gaussian distribution. Let $f(\cdot) : \mathcal{X} \mapsto \mathbb{R}$ be a random function, and denote the density of GP as $\mathrm{GP}(f(\cdot); \nu(\cdot), k(\cdot, \cdot))$, where $\nu(\cdot)$ and $k(\cdot, \cdot)$ are the mean function and the kernel (covariance) function, respectively. Intuitively, stating that a random function $f(\cdot)$ follows GP, i.e., $p(f(\cdot)) = \mathrm{GP}(f(\cdot); \nu(\cdot), k(\cdot, \cdot))$, means that the function values $f(\boldsymbol{x})$ indexed by the continuous input variable $\boldsymbol{x} \in \mathcal{X}$ follow the infinite-dimensional version (i.e., process) of the Gaussian distribution. The marginalization property [32] of the Gaussian allows the following definition:

**Definition 1.** *(Gaussian process)  GP is the process of a random function such that, if $f(\cdot) \sim \mathrm{GP}(\nu(\cdot), k(\cdot, \cdot))$, then for any set of input points $\boldsymbol{X} = (\boldsymbol{x}_1, \ldots, \boldsymbol{x}_N) \in \mathcal{X}^N$ it holds that*

$$p(\boldsymbol{f}|\boldsymbol{\nu}, \boldsymbol{K}) = \mathcal{N}_D(\boldsymbol{f}; \boldsymbol{\nu}, \boldsymbol{K}), \tag{14}$$

*where*

$$\boldsymbol{f} = (f(\boldsymbol{x}_1), \dots, f(\boldsymbol{x}_N))^\top \in \mathbb{R}^N, \qquad \boldsymbol{\nu} = (\nu(\boldsymbol{x}_1), \dots, \nu(\boldsymbol{x}_N))^\top \in \mathbb{R}^N,$$

$$\boldsymbol{K} = \begin{pmatrix} k(\boldsymbol{x}_1, \boldsymbol{x}_1) & \cdots & k(\boldsymbol{x}_1, \boldsymbol{x}_N) \\ \vdots & & \vdots \\ k(\boldsymbol{x}_N, \boldsymbol{x}_1) & \cdots & k(\boldsymbol{x}_N, \boldsymbol{x}_N) \end{pmatrix} \in \mathbb{R}^{N \times N}.$$

Consider another set of input points $\boldsymbol{X}' = (\boldsymbol{x}'_1, \dots, \boldsymbol{x}'_M) \in \mathcal{X}^M$, and let $\boldsymbol{f}' = (f(\boldsymbol{x}'_1), \dots, f(\boldsymbol{x}'_N))^\top \in \mathbb{R}^M, \boldsymbol{\nu}' = (\nu(\boldsymbol{x}'_1), \dots, \nu(\boldsymbol{x}'_N))^\top \in \mathbb{R}^M$ be the corresponding random function values and the mean function values, respectively. Then, Definition 1 implies that the joint distribution of $\boldsymbol{f}$ and $\boldsymbol{f}'$ is

$$p(\boldsymbol{f}, \boldsymbol{f}') = \mathcal{N}_{N+M}(\widetilde{\boldsymbol{f}}; \widetilde{\boldsymbol{\nu}}, \widetilde{\boldsymbol{K}}), \tag{15}$$

where

$$\widetilde{\boldsymbol{f}} = (\boldsymbol{f}^\top, \boldsymbol{f}'^\top)^\top \in \mathbb{R}^{N+M}, \qquad \widetilde{\boldsymbol{\nu}} = (\boldsymbol{\nu}^\top, \boldsymbol{\nu}'^\top)^\top \in \mathbb{R}^{N+M},$$

$$\widetilde{\boldsymbol{K}} = \begin{pmatrix} \boldsymbol{K} & \boldsymbol{K}' \\ \boldsymbol{K}'^\top & \boldsymbol{K}'' \end{pmatrix} \in \mathbb{R}^{(N+M) \times (N+M)}.$$

Here, $\boldsymbol{K} = k(\boldsymbol{X}, \boldsymbol{X}) \in \mathbb{R}^{N \times N}, \boldsymbol{K}' = k(\boldsymbol{X}, \boldsymbol{X}') \in \mathbb{R}^{N \times M}$, and $\boldsymbol{K}'' = k(\boldsymbol{X}', \boldsymbol{X}') \in \mathbb{R}^{M \times M}$, where $k(\boldsymbol{X}, \boldsymbol{X}')$ denotes the kernel matrix evaluated at each column of $\boldsymbol{X}$ and $\boldsymbol{X}'$ such that $(k(\boldsymbol{X}, \boldsymbol{X}'))_{n,m} = k(\boldsymbol{x}_n, \boldsymbol{x}_m)$.

The conditional distribution of $\boldsymbol{f}'$ given $\boldsymbol{f}$ can be analytically derived as

$$p(\boldsymbol{f}'|\boldsymbol{f}) = \mathcal{N}_M(\boldsymbol{f}'; \boldsymbol{\mu}_{\mathrm{cond}}, \boldsymbol{S}_{\mathrm{cond}}), \tag{16}$$

where

$$\boldsymbol{\mu}_{\mathrm{cond}} = \boldsymbol{\nu}' + \boldsymbol{K}'^\top \boldsymbol{K}^{-1}(\boldsymbol{f} - \boldsymbol{\nu}) \in \mathbb{R}^M, \qquad \boldsymbol{S}_{\mathrm{cond}} = \boldsymbol{K}'' - \boldsymbol{K}'^\top \boldsymbol{K}^{-1}\boldsymbol{K}' \in \mathbb{R}^M.$$

In GP regression, $\boldsymbol{X}$ and $\boldsymbol{X}'$ correspond to the training and the test inputs, respectively. The basic idea is to use the joint distribution (15) as the prior distribution on the training and the test points, and transform the likelihood information from $\boldsymbol{f}$ to $\boldsymbol{f}'$ by using the conditional (16).

The GP regression model consists of the Gaussian noise likelihood and GP prior:

$$p(y|\boldsymbol{x}, f(\cdot)) = \mathcal{N}_1(y; f(\boldsymbol{x}), \sigma^2), \qquad p(f(\cdot)) = \mathrm{GP}(f(\cdot); \nu(\cdot), k(\cdot, \cdot)), \tag{17}$$

where $\sigma^2$ denotes the observation noise variance. Below, we assume that the prior mean function is the constant zero function, i.e., $\nu(\boldsymbol{x}) = 0, \forall \boldsymbol{x}$. Derivations for the general case can be obtained by re-defining the observation and the random function as $y \leftarrow y - \nu(\boldsymbol{x})$ and $f(\boldsymbol{x}) \leftarrow f(\boldsymbol{x}) - \nu(\boldsymbol{x})$, respectively.

Given the training inputs and outputs, $\boldsymbol{X} = (\boldsymbol{x}_1, \dots, \boldsymbol{x}_N) \in \mathcal{X}^N$ and $\boldsymbol{y} = (y_1, \dots, y_N)^T \in \mathbb{R}^N$, the posterior of the function values at the training input points $\boldsymbol{f} = (f(\boldsymbol{x}_1), \dots, f(\boldsymbol{x}_N))^\top \in \mathbb{R}^N$ is given as

$$p(\boldsymbol{f}|\boldsymbol{X}, \boldsymbol{y}) = \frac{p(\boldsymbol{y}|\boldsymbol{X}, \boldsymbol{f})p(\boldsymbol{f})}{p(\boldsymbol{y}|\boldsymbol{X})} = \mathcal{N}_N(\boldsymbol{f}; \boldsymbol{\mu}, \boldsymbol{S}), \tag{18}$$

where

$$\boldsymbol{\mu} = \sigma^{-2}\left(\boldsymbol{K}^{-1} + \sigma^{-2}\boldsymbol{I}_N\right)^{-1}\boldsymbol{y} \in \mathbb{R}^N, \qquad \boldsymbol{S} = \left(\boldsymbol{K}^{-1} + \sigma^{-2}\boldsymbol{I}_N\right)^{-1} \in \mathbb{R}^{N \times N}.$$

Given the test inputs $\boldsymbol{X}' = (\boldsymbol{x}'_1, \dots, \boldsymbol{x}'_M) \in \mathcal{X}^M$, the posterior of the function values at the test points $\boldsymbol{f}' = (f(\boldsymbol{x}'_1), \dots, f(\boldsymbol{x}'_M))^\top \in \mathbb{R}^M$ can be obtained, by using Eqs. (16) and (18), as

$$p(\boldsymbol{f}'|\boldsymbol{X}, \boldsymbol{y}) = \int p(\boldsymbol{f}'|\boldsymbol{f})p(\boldsymbol{f}|\boldsymbol{X}, \boldsymbol{y}) = \mathcal{N}_M(\boldsymbol{f}'; \boldsymbol{\mu}', \boldsymbol{S}'), \tag{19}$$

where

$$\boldsymbol{\mu}' = \boldsymbol{K}'^\top\left(\boldsymbol{K} + \sigma^2\boldsymbol{I}_N\right)^{-1}\boldsymbol{y} \in \mathbb{R}^M, \quad \boldsymbol{S}' = \boldsymbol{K}'' - \boldsymbol{K}'^\top\left(\boldsymbol{K} + \sigma^2\boldsymbol{I}_N\right)^{-1}\boldsymbol{K}' \in \mathbb{R}^{M \times M}. \tag{20}$$

The predictive distribution of the output $\boldsymbol{y}' = (f(\boldsymbol{x}'_1) + \varepsilon'_1, \ldots, f(\boldsymbol{x}'_M) + \varepsilon'_M)^\top \in \mathbb{R}^M$ is given as

$$p(\boldsymbol{y}'|\boldsymbol{X}, \boldsymbol{y}) = \int p(\boldsymbol{y}'|\boldsymbol{f}')p(\boldsymbol{f}'|\boldsymbol{X}, \boldsymbol{y})d\boldsymbol{f}' = \mathcal{N}_M(\boldsymbol{y}'; \boldsymbol{\mu}'_y, \boldsymbol{S}'_y), \tag{21}$$

where

$$\boldsymbol{\mu}'_y = \boldsymbol{\mu}' \in \mathbb{R}^M, \qquad \boldsymbol{S}'_y = \boldsymbol{S}' + \sigma^2 \boldsymbol{I}_N \in \mathbb{R}^{M \times M}.$$

The marginal distribution of the training outputs is also analytically derived:

$$p(\boldsymbol{y}|\boldsymbol{X}) = \int p(\boldsymbol{y}|\boldsymbol{X}, \boldsymbol{f})p(\boldsymbol{f})d\boldsymbol{f} = \mathcal{N}_N(\boldsymbol{y}; \boldsymbol{\mu}_{\text{marg}}, \boldsymbol{S}_{\text{marg}}), \tag{22}$$

where

$$\boldsymbol{\mu}_{\text{marg}} = \boldsymbol{0} \in \mathbb{R}^N, \qquad \boldsymbol{S}_{\text{marg}} = \sigma^2 \boldsymbol{I}_N + \boldsymbol{K} \in \mathbb{R}^{N \times N}.$$

The marginal likelihood (22) is used for hyperparameter optimization.

## B.2 Bayesian Optimization

In BO [14], a surrogate function, which in most cases is GP regression, equipped with uncertainty estimation, is learned from the currently available observations. A new set of points that likely improves the current best score is observed in each iteration. Assume that at the $t$-th iteration of BO, we have already observed $N$ points $\boldsymbol{X}^{t-1} \in \mathcal{X}^N$. BO suggests a new set of $M$ points $\boldsymbol{X}' \in \mathcal{X}^M$ by solving the following problem:

$$\max_{\boldsymbol{X}'} a_{\boldsymbol{X}^{t-1}}(\boldsymbol{X}'),$$

where $a_{\boldsymbol{X}}(\cdot)$ is an *acquisition function* computed based on the GP trained on the observations $\boldsymbol{y}$ at $\boldsymbol{X}$. A popular choice for the acquisition function is Expected Improvement (EI) [27, 34],

$$a_{\boldsymbol{X}}^{\text{EI}}(\boldsymbol{x}') = \left\langle \max(0, \underline{f} - f') \right\rangle_{p(f'|\boldsymbol{X}, \boldsymbol{y})},$$

which covers the case where the observation noise is negligible and only a single point $\boldsymbol{x}'$ is chosen in each iteration, i.e., $\sigma^2 \ll 1, M = 1$. Here, $\underline{f}$ denotes the current best observation, i.e., $\underline{f} = \min_{n \in \{1, \ldots, N\}} f(\boldsymbol{x}_n)$, $\langle \cdot \rangle_p$ denotes the expectation value with respect to the distribution $p$, and $p(f'|\boldsymbol{X}, \boldsymbol{y})$ is the posterior distribution (19) of the function value $f'$ at the new point $\boldsymbol{x}'$. EI can be analytically computed:

$$a_{\boldsymbol{X}}^{\text{EI}}(\boldsymbol{x}') = (\underline{f} - \mu'_{\boldsymbol{X}})\Phi\left(\frac{\underline{f} - \mu'_{\boldsymbol{X}}}{s'_{\boldsymbol{X}}}\right) + s'_{\boldsymbol{X}}\phi\left(\frac{\underline{f} - \mu'_{\boldsymbol{X}}}{s'_{\boldsymbol{X}}}\right), \tag{23}$$

where $\mu'_{\boldsymbol{X}} \in \mathbb{R}$ and $s'_{\boldsymbol{X}} \in \mathbb{R}$ are the GP posterior mean and variance (20) with their subscripts indicating the input points on which the GP was trained, and

$$\phi(\varepsilon) = \mathcal{N}_1(\varepsilon; 0, 1^2), \qquad\qquad \Phi(\varepsilon) = \int_{-\infty}^{\varepsilon} \mathcal{N}_1(\varepsilon; 0, 1^2)d\varepsilon,$$

are the probability density function (PDF) and the cumulative distribution function (CDF), respectively, of the one-dimensional standard Gaussian.

For the general case where $\sigma^2 > 0, M \geq 1$, Noisy Expected Improvement (NEI) [18, 31] was proposed:

$$a_{\boldsymbol{X}}^{\text{NEI}}(\boldsymbol{X}') = \left\langle \max\left(0, \min(\boldsymbol{f}) - \min(\boldsymbol{f}')\right) \right\rangle_{p(\boldsymbol{f}, \boldsymbol{f}'|\boldsymbol{X}, \boldsymbol{y})}. \tag{24}$$

NEI treats the function values $\boldsymbol{f}$ at the already observed points $\boldsymbol{X}$ still as random variables, and appropriately takes the correlations between all pairs of old and new points into account. This is beneficial because it can avoid overestimating the expected improvements at, e.g., points close to the current best point and a set of two close new points in promising regions. A downside is that NEI does not have an analytic form, and requires quasi-Monte Carlo sampling for estimation. Moreover, maximization is not straightforward, and an approximate solution is obtained by sequentially adding a point to $\boldsymbol{X}'$ until $M$ points are collected.

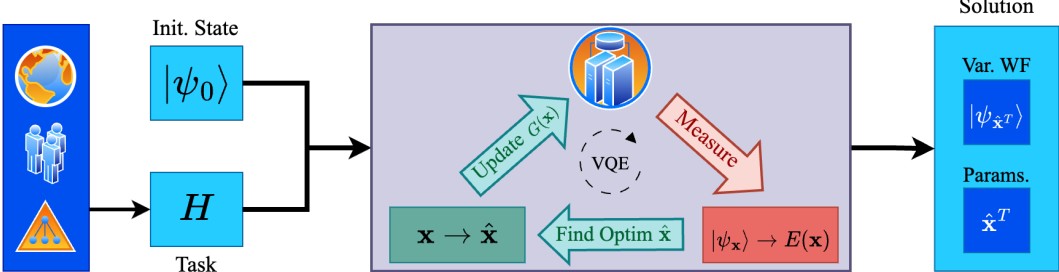

Figure 5: Illustration of the VQE workflow. In the first step, highly complicated optimization problems can be translated into the Hamiltonian formulation (see, e.g., [50, 51]). The Hamiltonian $H$ and an initial state $|\psi_0\rangle$ are plugged into the VQE block (light purple), where the variational quantum circuit is instantiated with random angular parameters $\mathbf{x}^0$. In the VQE block, the top vertex of the triangle represents the quantum computer, the red arrows and blocks refer to operations running on a quantum computer, while the green parts refer to classical steps. In the bottom-left green box, the current parameters $\mathbf{x}$ are updated with the new best parameters $\hat{\mathbf{x}}$ found during classical optimization routines. Then, the quantum circuit $G(\mathbf{x})$ is updated using the new optimum point, $\mathbf{x} \rightarrow \hat{\mathbf{x}}$, and the energy $E(\mathbf{x})$ for the updated variational wave function $|\psi_{\mathbf{x}}\rangle$ is measured. The VQE block is executed for $T$ iterations and finally outputs the solution to the task as a variational approximation $|\psi_{\hat{\mathbf{x}}^T}\rangle$ of the ground state and the corresponding optimal parameters $\hat{\mathbf{x}}^T$ for the quantum circuit.

Our EMICoRe, proposed in Section 3.2, can be seen as a generalization of NEI. In EMICoRe, the points in the confident regions (CoRe), where the predictive uncertainty after new points would have been observed is lower than a threshold $\kappa$, are treated as "indirectly observed", and the best score is searched for over CoRe. If we replace CoRe with the previous and the new training points, EMICoRe reduces to NEI.

Another related method to our approach is Knowledge Gradient (KG) [30]:

$$a_{\boldsymbol{X}}^{\text{KG}}(\boldsymbol{X}') = \left\langle \max\left(0, \min_{\boldsymbol{x}'' \in \mathcal{X}} \mu_{\boldsymbol{X}}(\boldsymbol{x}'') - \min_{\boldsymbol{x}'' \in \mathcal{X}} \mu_{(\boldsymbol{X}, \boldsymbol{X}')}(\boldsymbol{x}'')\right) \right\rangle_{p(\boldsymbol{y}'|\boldsymbol{X}, \boldsymbol{y})}, \qquad (25)$$

which assumes that the minimizer of the GP posterior mean function is used as the best score—even if the uncertainty at the minimizer is high—and estimates the improvement of the best score before and after the new points $\boldsymbol{X}'$ are added to the training data. The second term in Eq. (25) is estimated by simulation: it trains the GP on the augmented data $(\boldsymbol{X}, \boldsymbol{X}')$ and $(\boldsymbol{y}^\top, \boldsymbol{y}'^\top)^\top$, where $\boldsymbol{y}'$ are drawn from the GP posterior $p(\boldsymbol{y}'|\boldsymbol{X}, \boldsymbol{y})$, and finds the minimizer of the updated GP mean. Iterating this process provides Monte Carlo samples to estimate the second term in Eq. (25).

In our EMICoRe method, if we set the CoRe threshold $\kappa^2 \rightarrow \infty$ so that the entire search domain is in CoRe, and replace the random function $f(\cdot)$ in Eq. (11) with its previous and updated GP means, respectively, EMICoRe reduces to KG. Thus, KG can be seen as a version of EMICoRe that ignores the uncertainty of the updated GP.

## C  Details of Variational Quantum Eigensolvers (VQEs)

The VQE [7, 8] is a hybrid quantum-classical algorithm that uses a classical optimization routine in conjunction with a quantum processor to approximate the ground state of a given Hamiltonian. VQEs are designed to run on Noisy Intermediate-Scale Quantum (NISQ) devices, which are the current generation of quantum computers. These devices have a limited number of qubits and high error rates, which makes it challenging to execute complex quantum algorithms faithfully on these quantum devices. VQEs are a promising approach to tackle this challenge since they use a hybrid classical-quantum algorithm, where the quantum device only needs to perform relatively simple computations (see Fig. 5 for an illustration of the VQE workflow).

VQEs are considered to be potentially relevant for some challenging problems in different scientific domains such as quantum chemistry [52, 53, 54], drug discovery [55, 56, 57], condensed matter physics [58], materials science [59] and quantum field theories [60, 61, 62, 41]. Specifically, finding

the ground state of a molecule is a very challenging problem growing exponentially in complexity with the number of atoms for classical computing, while for VQE this would instead scale polynomially. Despite being naturally designed to solve problems associated with quantum chemistry and physics, such as calculating molecular energies, optimizing molecular geometries [63], and simulating strongly correlated systems [64], VQEs have also been applied to other domains including optimization and combinatorial problems such as the flight gate assignment [50, 51].

Given a Hamiltonian formulation that can be efficiently measured on a quantum device, the variational approach of VQE can be applied to obtain an upper bound for the ground-state energy as well as an approximation for the ground-state wave function. Let $|\psi_0\rangle$ be the initial ansatz for the $Q$-dimensional (qubit) wave function. Let us assume that we use a parametrized quantum circuit $G(\boldsymbol{x})$, where $\boldsymbol{x} \in [0, 2\pi)^D$ represents the angular parameters of quantum gates. The circuit $G$ consists of $D'(\geq D)$ unitary gates:

$$G(\boldsymbol{x}) = G_{D'} \circ \cdots \circ G_1, \tag{26}$$

where $D$ of the $D'$ gates depend on one of the angular parameters exclusively, i.e., $x_d$ parametrizes only a single gate $G_{d'(d)}(x_d)$, where $d'(d)$ specifies the gate parametrized by $x_d$.[VII] We consider the parametric gates of the form

$$G_{d'}(x) = U_{d'}(x) = \exp\left\{-i\frac{x}{2}P_{d'}\right\}, \tag{27}$$

where $P_{d'}$ is an arbitrary sequence of the Pauli operators $\{\sigma_q^X, \sigma_q^Y, \sigma_q^Z\}_{q=1}^Q$ acting on each qubit at most once. This form covers not only single-qubit gates such as $R_X(x) = \exp\left(-i\theta\sigma_q^X\right)$, but also entangling gates such as $R_{XX}(x) = \exp\left(-ix\sigma_{q_1}^X \circ \sigma_{q_2}^X\right)$ and $R_{ZZ}(x) = \exp\left(-ix\sigma_{q_1}^Z \circ \sigma_{q_2}^Z\right)$ for $q_1 \neq q_2$. In the matrix representation of quantum mechanics, quantum states are expressed as vectors in the computational basis, i.e.,

$$\langle 0| = (1 \quad 0), \qquad |0\rangle = \begin{pmatrix} 1 \\ 0 \end{pmatrix}, \qquad \langle 1| = (0 \quad 1), \qquad |1\rangle = \begin{pmatrix} 0 \\ 1 \end{pmatrix}. \tag{28}$$

Moreover, Pauli operators are expressed as matrices,

$$\sigma^X = \begin{pmatrix} 0 & 1 \\ 1 & 0 \end{pmatrix}, \qquad \sigma^Y = \begin{pmatrix} 0 & -i \\ i & 0 \end{pmatrix}, \qquad \sigma^Z = \begin{pmatrix} 1 & 0 \\ 0 & -1 \end{pmatrix},$$

acting only on one of the $Q$ qubits non-trivially. The application of any operator on a quantum state $|\psi\rangle$ thus becomes a matrix multiplication in the chosen basis. Single-qubit parametric gates correspond to rotations around the axes in a three-dimensional space representation of a qubit state, known as the Bloch sphere, and are widely used in ansatz circuits for VQEs.

Given the Hamiltonian $H$, which is an Hermitian operator, the quantum device measures the energy of the resulting quantum state $|\psi_{\boldsymbol{x}}\rangle = G(\boldsymbol{x})|\psi_0\rangle$ contaminated with observation noise $\varepsilon$, i.e.,

$$f(\boldsymbol{x}) = f^*(\boldsymbol{x}) + \varepsilon, \quad \text{where} \quad f^*(\boldsymbol{x}) = \langle\psi_{\boldsymbol{x}}|H|\psi_{\boldsymbol{x}}\rangle = \langle\psi_0|G(\boldsymbol{x})^\dagger H G(\boldsymbol{x})|\psi_0\rangle. \tag{29}$$

The observation noise $\varepsilon$ in our numerical experiments only incorporates the shot noise coming from the intrinsically probabilistic nature of quantum measurements, and the errors from imperfect qubits, gates, and measurements on current NISQ devices are not considered.

The task of the classical computer in VQE is to find the optimal angular parameters $\boldsymbol{x}$ such that $f^*(\boldsymbol{x})$ is minimal. Given that the ansatz circuit is expressive enough, $G(\boldsymbol{x})|\psi_0\rangle$ then corresponds to the ground state of the Hamiltonian $H$. Thus, the problem to be solved is a noisy black-box optimization:

$$\min_{\boldsymbol{x}\in[0,2\pi)^D} f^*(\boldsymbol{x}).$$

The long-term goal of research on VQEs is to develop efficient quantum algorithms that can solve problems beyond the capabilities of classical computers. While VQE is a promising approach for exploiting the advantages of quantum computing, they are plagued by some limitations. Specifically, these algorithms require appropriate choices for the quantum circuit [65, 66]. To have compact circuits with a moderate number of quantum gates is, therefore, essential to favor stable computations and high measurement accuracy.

---

[VII]In Appendix E, we discuss how the theorems and our method can be extended to the non-exclusive parametrization case.

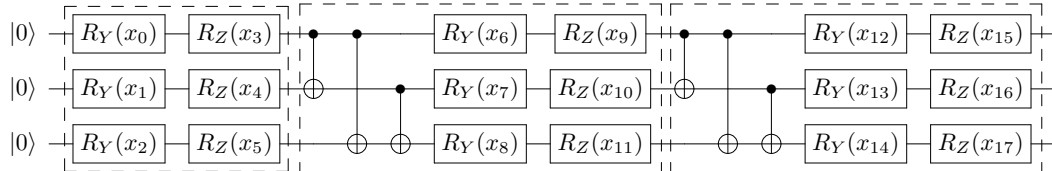

Figure 6: Illustration of Qiskit's [43] `Efficient SU(2) Circuit` with default parameters for $Q = 3$ qubits and $L = 2$ layers (thus $\boldsymbol{x} \in [0, 2\pi)^{18}$). Each dashed box indicates a layer of the ansatz circuit. The quantum computation proceeds from left to right with an initial ansatz of the form $|0\rangle^{\otimes Q}$. Each horizontal line corresponds to a quantum wire representing one qubit. The symbols on the wires correspond to gate operations on the respective qubits, starting with two parametrized rotational gates $R_Y$ and $R_Z$ acting on the qubits in the initial 0-th layer. Then, each of the following layers is composed of one block of `CNOT` gates and two rotational gates acting on each qubit.

**Efficient SU(2) circuit:** One circuit ansatz commonly used is the `Efficient SU(2) Circuit` from Qiskit [43], which is illustrated in Figure 6. In this circuit, two consecutive rotational gates,

$$R_Y(x) = \exp\left\{-i\frac{x}{2}\sigma^Y\right\} \qquad \text{and} \qquad R_Z(x) = \exp\left\{-i\frac{x}{2}\sigma^Z\right\}, \tag{30}$$

act on each qubit in the initial (0-th) layer. The rest of the circuit is composed by a stack of $L$ layers, each of which consists of `CNOT` gates applied to all pairs of qubits, and a pair of the rotational gates, $R_Y$ and $R_Z$, acting on each qubit. The `CNOT` gates are not parametrized and are therefore not updated during the optimization process. Therefore, the total number of angular parameters of the circuit is equal to the number of $R_Y$ and $R_Z$ gates in the circuit. In total, for a setup having $Q$ qubits and $L$ layers, the number of angular parameters is

$$D = 2 \times Q + (L \times 2) \times Q, \tag{31}$$

where the first term counts the rotational gates in the initial layer, while the second term counts those in the latter layers. In our experiments, we set the initial ansatz to $|\psi_0\rangle = |0\rangle^{\otimes Q}$, which corresponds to the tensor product of $Q$ qubits in the $|0\rangle$-ket state.[VIII]

## D  Proof of Theorem 1

*Proof.* The VQE kernel (9) can be rewritten as

$$k^{\text{VQE}}(\boldsymbol{x}, \boldsymbol{x}') = \sigma_0^2(\gamma^2 + 2)^{-D} \prod_{d=1}^{D} \left(\gamma^2 + 2\cos(x_d - x_d')\right)$$

$$= \sigma_0^2(\gamma^2 + 2)^{-D} \prod_{d=1}^{D} \left(\gamma^2 + 2\cos x_d \cos x_d' + 2\sin x_d \sin x_d'\right)$$

$$= \sigma_0^2(\gamma^2 + 2)^{-D} \sum_{\boldsymbol{\xi} \in \{0,1,2\}^D} \prod_{d=1}^{D} (\gamma^2)^{\mathbb{1}(\xi_d=0)} \left(2\cos x_d \cos x_d'\right)^{\mathbb{1}(\xi_d=1)} \left(2\sin x_d \sin x_d'\right)^{\mathbb{1}(\xi_d=2)}$$

$$= \boldsymbol{\phi}(\boldsymbol{x})^\top \boldsymbol{\phi}(\boldsymbol{x}'),$$

where $\mathbb{1}(\cdot)$ denotes the indicator function (equal to one if the event is true and zero otherwise), and

$$\boldsymbol{\phi}(\boldsymbol{x}) = \sigma_0(\gamma^2 + 2)^{-D/2} \cdot \mathbf{vec}\left(\otimes_{d=1}^{D}(\gamma, \sqrt{2}\cos x_d, \sqrt{2}\sin x_d)^\top\right) \in \mathbb{R}^{3^D},$$

which completes the proof. $\qquad \square$

## E  Generalization to Non-Exclusive Parametrization Case

In the VQE, it is often beneficial to share the same parameter amongst multiple gates, for example, in the case where the Hamiltonian has a certain symmetry, such as translation invariance. Doing so,

---

[VIII]An equivalent notation often found in the literature is $|0\rangle^{\otimes Q} = |\mathbf{0}\rangle$.

the variational quantum circuit is guaranteed to generate quantum states that fulfill the symmetry, and thus the number of parameters to be optimized is efficiently reduced. To consider this case, we need to assume that some of the entries of the search variable $\boldsymbol{x}$ are shared parameters, each of which parametrizes multiple gates. The Nakanishi-Fujii-Todo (NFT) algorithm [11] can still be used in this case, based on the following generalization of Proposition 2:

**Proposition 3.** *[11] Assume that the d-th entry of the input $\boldsymbol{x} \in [0, 2\pi)^D$ parametrizes $V_d \geq 1$ gate parameters. Then, for the VQE objective function $f^*(\cdot)$ in Eq. (6),*

$$\exists \boldsymbol{b} \in \mathbb{R}^{\Pi_{d=1}^D (1+2V_d)} \quad such \ that \quad f^*(\boldsymbol{x}) = \boldsymbol{b}^\top \cdot \mathbf{vec}\left( \otimes_{d=1}^D \begin{pmatrix} 1 \\ \cos x_d \\ \vdots \\ \cos(V_d x_d) \\ \sin x_d \\ \vdots \\ \sin(V_d x_d) \end{pmatrix} \right). \tag{32}$$

Similarly, our VQE kernel introduced in Theorem 1 can be generalized as follows:

**Theorem 3.** *The (higher-order) VQE kernel,*

$$k^{VQE}(\boldsymbol{x}, \boldsymbol{x}') = \sigma_0^2 \prod_{d=1}^D \left( \frac{\gamma^2 + 2\sum_{v=1}^{V_d} \cos\left(v(x_d - x_d')\right)}{\gamma^2 + 2V_d} \right), \tag{33}$$

*is decomposed as $k^{VQE}(\boldsymbol{x}, \boldsymbol{x}') = \boldsymbol{\phi}(\boldsymbol{x})^\top \boldsymbol{\phi}(\boldsymbol{x}')$, where*

$$\boldsymbol{\phi}(\boldsymbol{x}) = \sigma_0 \left(\gamma^2 + 2V_d\right)^{-D/2} \cdot \mathbf{vec}\left( \otimes_{d=1}^D \begin{pmatrix} \gamma \\ \sqrt{2}\cos x_d \\ \vdots \\ \sqrt{2}\cos(V_d x_d) \\ \sqrt{2}\sin x_d \\ \vdots \\ \sqrt{2}\sin(V_d x_d) \end{pmatrix} \right). \tag{34}$$

*Proof.* Similarly to the *exclusive parameterization* (or first-order) case, we have

$$k^{\mathrm{VQE}}(\boldsymbol{x}, \boldsymbol{x}') = \sigma_0^2 \prod_{d=1}^D \left( \frac{\gamma^2 + 2\sum_{v=1}^{V_d} \cos\left(v(x_d - x_d')\right)}{\gamma^2 + 2V_d} \right)$$

$$= \sigma_0^2 \left(\gamma^2 + 2V_d\right)^{-D} \prod_{d=1}^D \left( \gamma^2 + 2\sum_{v=1}^{V_d} \{\cos(vx_d)\cos(vx_d') + \sin(vx_d)\sin(vx_d')\} \right)$$

$$= \sigma_0^2 \left(\gamma^2 + 2V_d\right)^{-D}$$

$$\sum_{\boldsymbol{\xi} \in \{0,\ldots,2V_d\}^D} \prod_{d=1}^D (\gamma^2)^{\mathbb{1}(\xi_d=0)} \prod_{v=1}^{V_d} \left(2\cos(vx_d)\cos(vx_d')\right)^{\mathbb{1}(\xi_d=2v-1)} \left(2\sin(vx_d)\sin(vx_d')\right)^{\mathbb{1}(\xi_d=2v)}$$

$$= \boldsymbol{\phi}(\boldsymbol{x})^\top \boldsymbol{\phi}(\boldsymbol{x}'),$$

which completes the proof. $\qquad\square$

Since the VQE objective (32) is the $V_d$-th order sinusoidal function along the $x_d$-axis, NFT can perform the sequential minimal optimization (SMO) by observing $2V_d$ points in each step — which become $2V_d + 1$ points, together with the current optimum point — to determine the entire function form in the one-dimensional subspace parallel to the $x_d$-axis. Our NFT-with-EMICoRe approach can similarly be generalized to the non-exclusive cases: for the chosen direction $d$, the approach observes $M = 2V_d$ new points by maximizing the EMICoRe acquisition function, based on the GP regression with the generalized VQE kernel (33).

---

**Algorithm 1:** Nakanishi-Fuji-Todo (NFT) method [11] (Baseline)

---

**input :**
- $T_{\mathrm{MI}}$ : max # of iterations
- $T_{\mathrm{RI}}$ : reset interval
- $\mathcal{D}^0 = (\hat{\boldsymbol{x}}^0, \hat{y}^0)$ : initialization with $\hat{\boldsymbol{x}}^0 \sim [0, 2\pi)^D$ and $\hat{y}^0 = f^*(\hat{\boldsymbol{x}}^0) + \varepsilon$

**output :**
- $\hat{\boldsymbol{x}}^{T_{\mathrm{MI}}}$ : last optimal point
- $y(\hat{\boldsymbol{x}}^{T_{\mathrm{MI}}})$ : last estimated objective
- $\mathcal{D}^{T_{\mathrm{MI}}}$ : ensemble of collected observations

---

1 **begin**
2    **for** $t = 1$ **to** $T_{MI}$ **do**
3      Choose a direction $d \in \{1, \ldots, D\}$ sequentially or randomly;
4      `Find(`$\boldsymbol{X}'$`)` $\Longrightarrow$ $\boldsymbol{X}' = (\boldsymbol{x}'_1, \boldsymbol{x}'_2) = \{\hat{\boldsymbol{x}}^{t-1} - 2\pi/3\boldsymbol{e}_d, \hat{\boldsymbol{x}}^{t-1} + 2\pi/3\boldsymbol{e}_d\}$ along $d$;
5      `Observe` $\boldsymbol{y}'$ $\Longrightarrow$ $\boldsymbol{y}' = (y'_1, y'_2)^\top$ at the new points $\boldsymbol{X}'$;
6      `Append(`$\mathcal{D}^{t-1} \cup (\boldsymbol{X}', \boldsymbol{y}')$`)` $\Longrightarrow$ $\mathcal{D}^t$;
7      `Fit(`$\widetilde{f}(\theta)$`)` $\Longrightarrow$ $\widetilde{f}(\theta) = c_0 + c_1 \cos\theta + c_2 \sin\theta$ to the three points $\{(-2\pi/3, y'_1), (0, \hat{y}^{t-1}), (2\pi/3, y'_2)\}$ ;
8      `FindMin(`$\widetilde{f}$`)` $\Longrightarrow$ find analytical minimum $\hat{\theta} = \operatorname{argmin}_{\theta \in [0, 2\pi]} \widetilde{f}(\theta)$;
9      `Update` $\hat{\boldsymbol{x}}$ $\Longrightarrow$ with $\hat{\boldsymbol{x}}^t = \hat{\boldsymbol{x}}^{t-1} + \hat{\theta}\boldsymbol{e}_d$;
10      `Update(`$\hat{y}$`)` $\Longrightarrow$ with estimated $\hat{y}^t = \widetilde{f}(\hat{\theta})$;
11      **if** $t \mod T_{\mathrm{RI}} = 0$ **then**
12        `Observe` $y(\hat{\boldsymbol{x}}^t)$ ;        /* Perform additional observation */
13        $\mathcal{D}^t = \mathcal{D}^t \cup (\hat{\boldsymbol{x}}^t, y(\hat{\boldsymbol{x}}^t))$;
14      **end**
15    **end**
16 **end**
17 **return** $\mathcal{D}^{T_{\mathrm{MI}}}, \hat{\boldsymbol{x}}^{T_{\mathrm{MI}}}, y(\hat{\boldsymbol{x}}^{T_{\mathrm{MI}}})$

---

## F   Algorithm Details

Here, we provide the detailed procedures of the baseline method (Nakanishi-Fuji-Todo (NFT) [11]) and our proposed method (NFT-with-EMICoRe and the EMICoRe subroutine).

### F.1   Nakanishi-Fuji-Todo (NFT)

In each step of NFT (Algorithm 1), an axis $d \in \{1, \ldots, D\}$ is chosen sequentially or randomly, and the next observation points along the axis are set (Step 4) and observed (Step 5). Based on Proposition 2, the two new observations together with the previous optimum are fitted to a sinusoidal function, and the global optimum along the axis is analytically computed, establishing sequential minimal optimization (SMO) [12]. NFT iterates this process from a random initial point and outputs the collected datapoints and the last optimum. Since the optimal point at each step is not directly observed, errors can accumulate over iterations. As a remedy, NFT observes the optimal point when the reset interval condition is met (Step 12).

### F.2   NFT-with-EMICoRe

Our proposed method, NFT-with-EMICoRe (Algorithm 2), replaces the deterministic choice of the next observation points $\boldsymbol{X}'$ in NFT with a promising choice by BO. After initial (optional) iterations of NFT until sufficient training data are collected, we start the EMICoRe subroutine (Algorithm 3) to suggest new observation points $\boldsymbol{X}'$ by BO (Step 8). Then, the objective values $\boldsymbol{y}'$ at $\boldsymbol{X}'$ are observed (Step 10), and the training data $\mathcal{D}^{t-1} = \{\boldsymbol{X}^{t-1}, \boldsymbol{y}^{t-1}\}$ are updated with the new observed data

---

**Algorithm 2:** NFT-with-EMICoRe

**input :**
- $T_{\mathrm{MI}}$ : # of iterations
- $T_{\mathrm{NFT}}$ : # of initial NFT steps
- $T_{\mathrm{Ave}}(> T_{\mathrm{NFT}})$ : averaging steps for $\kappa$ update.
- $\mathcal{D}^0 = (\hat{\boldsymbol{x}}^0, \hat{y}^0)$ : initialization with $\hat{\boldsymbol{x}}^0 \sim [0, 2\pi)^D$ and $\hat{y}^0 = f^*(\hat{\boldsymbol{x}}^0) + \varepsilon$

**output :**
- $\hat{\boldsymbol{x}}^{T_{\mathrm{MI}}}$ : last optimal point
- $\mu(\hat{\boldsymbol{x}}^{T_{\mathrm{MI}}})$ : last estimated objective
- $\mathcal{D}^{T_{\mathrm{MI}}}$ : ensemble of collected observations

**1** **if** $T_{\mathrm{NFT}} > 0$ **then**
**2**     $\mathcal{D}^{T_{\mathrm{NFT}}}, \_, \_ = \mathrm{NFT}(T_{\mathrm{MI}} = T_{\mathrm{NFT}}, T_{\mathrm{RI}} = 0, D)$ ;        /* see Algorithm 1 */
**3**     Update observations $\mathcal{D}^{T_{\mathrm{NFT}}} = \mathcal{D} \cup \mathcal{D}^0$ ;      /* Collect points from NFT step */
**4** **end**

**5** **begin**
**6**     **for** $t = T_{NFT} + 1$ **to** $T_{MI}$ **do**
**7**        Choose a direction $d \in \{1, \ldots, D\}$ sequentially or randomly;
**8**        Find $\boldsymbol{X}' \implies$ points maximizing the $\mathrm{EMICoRe}(\hat{\boldsymbol{x}}^{t-1}, \mathcal{D}^{t-1}, d^t, \kappa^t)$ ;
**9**        /* for EMICoRe sub-routine see Algorithm 3                       */
**10**        Observe $\boldsymbol{y}' \implies \boldsymbol{y}' = (y_1', y_2')^\top$ at the new points $\boldsymbol{X}'$;
**11**        Append$(\mathcal{D}^{t-1} \cup (\boldsymbol{X}', \boldsymbol{y}')) \implies \mathcal{D}^t$ ;
**12**        Train $\mathrm{GP}(\mathcal{D}^t)$ on updated dataset;
**13**        Compute posterior means $\boldsymbol{\mu} = (\mu(\hat{\boldsymbol{x}}^{t-1} - 2\pi/3\boldsymbol{e}_d), \mu(\hat{\boldsymbol{x}}^{t-1}), \mu(\hat{\boldsymbol{x}}^{t-1} + 2\pi/3\boldsymbol{e}_d))^\top$;
**14**        Fit$(\widetilde{f}(\theta)) \implies \widetilde{f}(\theta) = c_0 + c_1 \cos\theta + c_2 \sin\theta$ to the three points
            $\{(-2\pi/3, \mu_1), (0, \mu_2), (2\pi/3, \mu_3)\}$ ;
**15**        FindMin$(\widetilde{f}) \implies$ find analytical min $\hat{\theta} = \mathrm{argmin}_{\theta \in [0, 2\pi]} \widetilde{f}(\theta)$ ;
**16**        Update $\hat{\boldsymbol{x}} \implies$ with $\hat{\boldsymbol{x}}^t = \hat{\boldsymbol{x}}^{t-1} + \hat{\theta}\boldsymbol{e}_d$;
**17**        Evaluate optimal objective $\hat{\mu}^t = \mu(\hat{x}^t)$;
**18**        **if** $t \geq T_{\mathrm{Ave}}$ **then**
**19**           Compute the CoRe threshold for the next iteration: $\kappa^{t+1} = \frac{\hat{\mu}^{t-T_{\mathrm{Ave}}} - \hat{\mu}^t}{T_{\mathrm{Ave}}}$;
**20**        **end**
**21**     **end**
**22** **end**
**23** **return** $\mathcal{D}^{T_{\mathrm{MI}}}, \hat{\boldsymbol{x}}^{T_{\mathrm{MI}}}, \mu(\hat{\boldsymbol{x}}^{T_{\mathrm{MI}}})$

---

$\{\boldsymbol{X}', \boldsymbol{y}'\}$ (Step 11). The GP is updated with the updated training data $\mathcal{D}^t = \{\boldsymbol{X}^t, \boldsymbol{y}^t\}$, and its mean predictions at three points are fitted by a sinusoidal function (Step 14) for finding the new optimal point $\hat{\boldsymbol{x}}^t$ (Step 15). Before going to the next iteration, the CoRe threshold $\kappa$ is updated according to the energy decrease in the last iterations (Step 19). In the early stage of the optimization, where the energy $\hat{\mu}^t$ decreases steeply, a large $\kappa$ encourages crude optimization, while in the converging phase where the energy decreases slowly, a small $\kappa$ enforces accurate optimization steps.

### F.3 EMICoRe Subroutine

The EMICoRe subroutine (Algorithm 3) receives the current optimal point $\hat{\boldsymbol{x}}$, the current training data $\{\boldsymbol{X}, \boldsymbol{y}\}$, the direction $d$ to be explored, and the CoRe threshold $\kappa$, and returns the suggested observation points. Fixing the number of new observations per step to $M = 2$, we prepare pairs of candidate points (sampled on a grid) along the axis $d$ as a candidate set $\mathcal{C} = \{\breve{\boldsymbol{X}}^j \in \mathbb{R}^{D \times 2}\}_{j=1}^{J_{\mathrm{SG}}(J_{\mathrm{SG}}-1)}$ (Step 2).

---

**Algorithm 3:** EMICoRe subroutine

**input** :
- $\hat{\boldsymbol{x}}$ : current optimal point
- $\mathcal{D} = \{\boldsymbol{X}, \boldsymbol{y}\}$ : current training data
- $d$ : direction to be explored
- $\kappa$ : CoRe threshold

**params** :
- $M = 2$ : # of suggested points
- $J_{\text{SG}}$ : # of search grid points
- $J_{\text{OG}}$ : # of evaluation grid points
- $N_{\text{MC}}$ : # of Monte Carlo samples

**output** :
- $X'$ : suggested observation points

1 **begin**
2      Prepare a candidate set $\mathcal{C} = \{\breve{\boldsymbol{X}}^j \in \mathbb{R}^{D \times 2}\}_{j=1}^{J_{\text{SG}}(J_{\text{SG}}-1)}$;
3      /* with $J_{\text{SG}}(J_{\text{SG}} - 1)$ being the # of candidate pairs             */
4      **for** $j = 1$ **to** $J_{\text{SG}}(J_{\text{SG}} - 1)$;            /* $j$ denotes one pair of points */
5      **do**
6          Update GP adding the current candidate point $\implies \text{GP}(\widetilde{\boldsymbol{X}})$ where $\widetilde{\boldsymbol{X}} = (\boldsymbol{X}, \breve{\boldsymbol{X}}^j)$
         Compute the posterior variance $s_{\widetilde{\boldsymbol{X}}}(\boldsymbol{x}, \boldsymbol{x}) \, \forall \boldsymbol{x}$ in the evaluation grid, along axis $d$ ;
7          /* test GP uncertainty on evaluation points.             */
8          Find discrete approximation of the CoRe as $\mathcal{Z}_{\widetilde{\boldsymbol{X}}} = \{\boldsymbol{x} \in \boldsymbol{X}^{\text{Grid}}; s_{\widetilde{\boldsymbol{X}}}(\boldsymbol{x}, \boldsymbol{x}) \leq \kappa^2\}$;
9          Use $\hat{\boldsymbol{x}}$ and $\boldsymbol{X}^{\text{test}} = \mathcal{Z}_{\widetilde{\boldsymbol{X}}}$ to compute *mean* and the *covariance* of GP posterior:
         $p(\hat{f}, \boldsymbol{f}^{\text{test}} | \boldsymbol{X}, \boldsymbol{y})$;
10         Estimate acquisition function by quasi Monte Carlo sampling
11         $a_{\boldsymbol{X}}^{\text{EMICoRe}}(j) = \frac{1}{M} \langle \max\{0, \hat{f} - \min(\boldsymbol{f}^{\text{test}})\} \rangle_{p(\hat{f}, \boldsymbol{f}^{\text{test}} | \boldsymbol{X}, \boldsymbol{y})}$ ;
12      **end**
13      Find pair of observation points that maximizes EMICoRe
14      $\hat{j} = \text{argmax}_j \, a_{\boldsymbol{X}}^{\text{EMICoRe}}(j)$ ;
15 **end**
16 **return** *Suggested points to observe at next step* $\boldsymbol{X}' = \breve{\boldsymbol{X}}^{\hat{j}}$

---

For each candidate pair, the predictive variances of the *updated* GP are computed on $\boldsymbol{x} \in \boldsymbol{X}^{\text{test}}$, points on a test grid, along the direction $d$ (Step 6). This way, a discrete approximation of the CoRe is obtained by collecting the grid points where the variance is smaller than the threshold $\kappa^2$ (Step 8). After computing the mean and the covariance of the current GP at the current optimum $\hat{\boldsymbol{x}}$ and on the (discrete) CoRe points (Step 9) — which is a $\widetilde{D}$-dimensional Gaussian for $\widetilde{D} = |\hat{\boldsymbol{x}} \cup \mathcal{Z}_{\widetilde{\boldsymbol{X}}}| = 1 + |\mathcal{Z}_{\widetilde{\boldsymbol{X}}}|$ — the EMICoRe acquisition function is estimated by quasi-Monte Carlo sampling (Step 11). After evaluating the acquisition functions for all candidate pairs of points, the best pair is returned as the suggested observation points (Step 16).

### F.4   Parameter Setting

We automatically tune the sensitive parameters: the kernel smoothness parameter $\gamma$ is optimized by maximizing the marginal likelihood in each iteration in the early phase of optimization, and at intervals in the later phase (see Appendix H for the concrete schedule in each experiment). The CoRe threshold $\kappa$ is updated at every step and set to the average energy decrease of the last iterations as in Step 19 of Algorithm 2, which performs comparably to the best heuristic in our investigation below, see Table 1. The noise variance $\sigma^2$ is set by observing $f^*(\boldsymbol{x})$ several times at several random

Table 1: Performance of EMICoRe depending on the choice of hyperparameters $C_0$ and $C_1$ for the CoRe threshold update rule (35). The best results are highlighted in bold, while $\downarrow$ ($\uparrow$) indicates whether lower (higher) values are better.

| Description | $C_0$ | $C_1$ | Energy $\downarrow$ | Fidelity $\uparrow$ |
|---|---|---|---|---|
| Default (Eq.(12)) | 0.0 | 1.0 | $-5.82 \pm 0.14$ | $0.85 \pm 0.16$ |
| Extreme (small) | 0.1 | 0.1 | $-5.82 \pm 0.11$ | $0.85 \pm 0.16$ |
| High (large) | 10.0 | 10.0 | $-5.72 \pm 0.15$ | $0.82 \pm 0.16$ |
| Extreme (large) | 10.0 | 100.0 | $-5.70 \pm 0.16$ | $0.80 \pm 0.18$ |
| **Best** | **0.1** | **10.0** | $\mathbf{-5.84 \pm 0.09}$ | $\mathbf{0.87 \pm 0.11}$ |

points and estimating $\sigma^2 = \hat{\sigma}^{*2}(N_{\text{shots}})$ before starting the optimization. For the GP prior, the zero mean function $\nu(\boldsymbol{x}) = 0, \forall \boldsymbol{x}$ is used, and the prior variance $\sigma_0^2$ is roughly set so that the standard deviation $\sigma_0$ is in the same order as the absolute value of the ground-state energy. The parameters $T_{\text{MI}}, J_{\text{SG}}, J_{\text{OG}}$, and $N_{\text{MC}}$ should be set to sufficiently large values as long as the (classical) computation is tractable, and the performance is not sensitive to $T_{\text{NFT}}$ and $T_{\text{Ave}}$. The parameter values used in our experiments are given in Appendix H.

**Investigation of CoRe Threshold Update Rules:** In our experiments in Section 4, the CoRe threshold is updated by Eq. (12). Here, we investigate whether a more fine-tuned update rule can improve the performance. Specifically, we test the following protocol:

$$\kappa^{t+1} = \max\left(C_0 \cdot \sigma, \ C_1 \cdot \frac{\hat{\mu}^{t-T_{\text{Ave}}} - \hat{\mu}^t}{T_{\text{Ave}}}\right), \tag{35}$$

where $C_0, C_1 \geq 0$ are the hyperparameters controlling the lower bound and the scaling of the average energy reduction, respectively. We note that $\sigma$ is the standard deviation of the observation noise, and setting the hyperparameters to $C_0 = 0$, $C_1 = 1.0$ reduces to the default update rule (12). Table 1 shows the achieved energy and fidelity for different values of the hyperparameters $C_0$ and $C_1$, after 600 observed points, in the setting of the Ising Hamiltonian at criticality and a $(L = 3)$-layered $(Q = 5)$-qubits quantum circuit with $N_{\text{shots}} = 1024$ readout shots. We observe that choosing $C_0 = 0.1$ and $C_1 = 10$ leads to the best performance; however, we also note that the setting used for the paper (12) achieves a similar performance.

## G  Proof of Theorem 2

*Proof.* We divide the proof into two steps.

### G.1  Eq. (7) $\Rightarrow$ Eq. (8)

The parameter shift rule (7) for $a = 1/2$ gives

$$2\frac{\partial}{\partial x_d} f^*(\boldsymbol{x}) = f^*\left(\boldsymbol{x} + \frac{\pi}{2}\boldsymbol{e}_d\right) - f^*\left(\boldsymbol{x} - \frac{\pi}{2}\boldsymbol{e}_d\right), \quad \forall \boldsymbol{x} \in [0, 2\pi)^D, d = 1, \dots, D, \tag{36}$$

which implies the differentiability of $f^*(\boldsymbol{x})$ in the whole domain $[0, 2\pi)^D$. For any $d = 1, \dots, D$ and $\hat{\boldsymbol{x}} \in [0, 2\pi)^D$, consider the one-dimensional subspace of the domain such that $\mathcal{A}_{d,\hat{\boldsymbol{x}}} = \{\hat{\boldsymbol{x}} + \alpha \boldsymbol{e}_d; \alpha \in [0, 2\pi)\}$, and the following restriction of $f^*(\cdot)$ on the subspace:

$$\widetilde{f}^*_{d,\hat{\boldsymbol{x}}}(x_d) \equiv f^*\big|_{\mathcal{A}_{d,\hat{\boldsymbol{x}}}}(\boldsymbol{x}) = f^*(\hat{\boldsymbol{x}} + (x_d - \hat{x}_d)\boldsymbol{e}_d).$$

For this restricted function, the parameter shift rule (36) applies as

$$2\frac{\partial}{\partial x_d}\widetilde{f}^*_{d,\hat{\boldsymbol{x}}}(x_d) = \widetilde{f}^*_{d,\hat{\boldsymbol{x}}}\left(x_d + \frac{\pi}{2}\right) - \widetilde{f}^*_{d,\hat{\boldsymbol{x}}}\left(x_d - \frac{\pi}{2}\right), \quad \forall x_d \in [0, 2\pi), d = 1, \dots, D. \tag{37}$$

The periodicity of $f^*(\cdot)$ requires that $\widetilde{f}^*_{d,\hat{\boldsymbol{x}}}(x_d)$ can be written as a Fourier series,

$$\widetilde{f}^*_{d,\hat{\boldsymbol{x}}}(x_d) = c_{d,0,0}(\hat{\boldsymbol{x}}_{\backslash d}) + \sum_{\tau=1}^{\infty}\left\{c_{d,1,\tau}(\hat{\boldsymbol{x}}_{\backslash d})\cos(\tau x_d) + c_{d,2,\tau}(\hat{\boldsymbol{x}}_{\backslash d})\sin(\tau x_d)\right\}, \tag{38}$$

where $\{c_{d,\cdot,\cdot}(\hat{\boldsymbol{x}}_{\backslash d})\}$ denote the Fourier coefficients, which depend on $\hat{\boldsymbol{x}}$ except for $\hat{x}_d$. Below, we omit the dependence of the Fourier coefficients on $\hat{\boldsymbol{x}}_{\backslash d}$ to avoid cluttering.

Substituting Eq. (38) into the left- and the right-hand sides of Eq. (37), respectively, gives

$$2\frac{\partial}{\partial x_d}\widetilde{f}^*_{d,\hat{\boldsymbol{x}}}(x_d) = 2\sum_{\tau=1}^{\infty}\tau\left(-c_{d,1,\tau}\sin\left(\tau x_d\right) + c_{d,2,\tau}\cos\left(\tau x_d\right)\right), \tag{39}$$

$$\widetilde{f}^*_{d,\hat{\boldsymbol{x}}}\left(x_d + \frac{\pi}{2}\right) - \widetilde{f}^*_{d,\hat{\boldsymbol{x}}}\left(x_d - \frac{\pi}{2}\right) = \sum_{\tau=1}^{\infty}\left(c_{d,1,\tau}\left\{\cos\left(\tau\left(x_d + \frac{\pi}{2}\right)\right) - \cos\left(\tau\left(x_d - \frac{\pi}{2}\right)\right)\right\}\right.$$

$$\left. + c_{d,2,\tau}\left\{\sin\left(\tau\left(x_d + \frac{\pi}{2}\right)\right) - \sin\left(\tau\left(x_d - \frac{\pi}{2}\right)\right)\right\}\right)$$

$$= 2\sum_{\tau=1}^{\infty}\left(-c_{d,1,\tau}\sin\left(\tau x_d\right)\sin\left(\frac{\tau\pi}{2}\right) + c_{d,2,\tau}\cos\left(\tau x_d\right)\sin\left(\frac{\tau\pi}{2}\right)\right)$$

$$= 2\sum_{\tau=1}^{\infty}\sin\left(\frac{\tau\pi}{2}\right)\left(-c_{d,1,\tau}\sin\left(\tau x_d\right) + c_{d,2,\tau}\cos\left(\tau x_d\right)\right). \tag{40}$$

Since Eq. (37) requires that Eqs. (39) and (40) are equal to each other for any $x_d \in [0, 2\pi)$ and $d = 1, \ldots, D$, it must hold that

$$\tau = \sin\left(\frac{\tau\pi}{2}\right) \quad \forall\tau \quad \text{such that } c_{d,1,\tau} \neq 0 \text{ or } c_{d,2,\tau} \neq 0. \tag{41}$$

Since Eq. (41) can hold only for $\tau = 1$, we deduce that

$$c_{d,1,\tau} = c_{d,2,\tau} = 0, \quad \forall\tau \neq 1, d = 1, \ldots, D.$$

Therefore, the restricted function must be the first-order sinusoidal function:

$$\widetilde{f}^*_{d,\hat{\boldsymbol{x}}}(x_d) = c_{d,0,0}(\hat{\boldsymbol{x}}_{\backslash d}) + c_{d,1,1}(\hat{\boldsymbol{x}}_{\backslash d})\cos\left(x_d\right) + c_{d,2,1}(\hat{\boldsymbol{x}}_{\backslash d})\sin\left(x_d\right). \tag{42}$$

As the most general function form that satisfies Eq. (42) for all $d = 1, \ldots, D$, we have

$$f^*(\boldsymbol{x}) = \sum_{\boldsymbol{\xi}\in\{0,1,2\}^D}\widetilde{b}_{\boldsymbol{\xi}}\prod_{d=1}^{D}1^{\mathbb{1}(\xi_d=0)}\cdot(\cos x_d)^{\mathbb{1}(\xi_d=1)}\cdot(\sin x_d)^{\mathbb{1}(\xi_d=2)}$$

$$= \boldsymbol{b}^{\top}\cdot\mathbf{vec}\left(\otimes_{d=1}^{D}(1, \cos x_d, \sin x_d)^{\top}\right).$$

Here, $\boldsymbol{\xi} \in \{0, 1, 2\}^D$ takes the value of either 0, 1, or 2, specifying the dependence on $x_d$—constant, cosine, or sine—for each entry, and $\widetilde{\boldsymbol{b}} = (\widetilde{b}_{\boldsymbol{\xi}})_{\boldsymbol{\xi}\in\{0,1,2\}^D}$ is the $3^D$-dimensional coefficient vector indexed by $\boldsymbol{\xi}$. With the appropriate bijective mapping $\iota : \{0, 1, 2\}^D \mapsto 1, \ldots, 3^D$ consistent with the definition of the vectorization operator $\mathbf{vec}(\cdot)$, we defined $\boldsymbol{b} \in \mathbb{R}^{3^D}$ such that $b_{\iota(\boldsymbol{\xi})} = \widetilde{b}_{\boldsymbol{\xi}}$. $\mathbb{1}(\cdot)$ is the indicator function, which is equal to one if the event is true and zero otherwise.

## G.2    Eq. (8) $\Rightarrow$ Eq. (7)

For any $f^*(\cdot)$ in the form of Eq. (8), the left- and right-hand sides of the parameter shift rule (36) can be, respectively, written as

$$2\frac{\partial}{\partial x_d}f^*(\boldsymbol{x}) = 2\left(-c_{d,1,1}(\boldsymbol{x}_{\backslash d})\sin\left(x_d\right) + c_{d,2,1}(\boldsymbol{x}_{\backslash d})\cos\left(x_d\right)\right), \tag{43}$$

$$f^*\left(\boldsymbol{x} + \frac{\pi}{2}\boldsymbol{e}_d\right) - f^*\left(\boldsymbol{x} - \frac{\pi}{2}\boldsymbol{e}_d\right) = c_{d,1,1}(\boldsymbol{x}_{\backslash d})\left\{\cos\left(x_d + \frac{\pi}{2}\right) - \cos\left(x_d - \frac{\pi}{2}\right)\right\}$$

$$+ c_{d,2,1}(\boldsymbol{x}_{\backslash d})\left\{\sin\left(x_d + \frac{\pi}{2}\right) - \sin\left(x_d - \frac{\pi}{2}\right)\right\}$$

$$= 2\left(-c_{d,1,1}(\boldsymbol{x}_{\backslash d})\sin\left(x_d\right)\sin\left(\frac{\pi}{2}\right) + c_{d,2,1}(\boldsymbol{x}_{\backslash d})\cos\left(x_d\right)\sin\left(\frac{\pi}{2}\right)\right)$$

$$= 2\left(-c_{d,1,1}(\boldsymbol{x}_{\backslash d})\sin\left(x_d\right) + c_{d,2,1}(\boldsymbol{x}_{\backslash d})\cos\left(x_d\right)\right), \tag{44}$$

which coincide with each other. This completes the proof. $\qquad\square$

Table 2: Choice of coupling parameters for the Ising and Heisenberg Hamiltonians for reproducing the experiments in Section 4 and Appendix I.

| | Ising | Heisenberg |
|---|---|---|
| `--j-couplings` $(J_X, J_Y, J_Z)$ | `(-1.0, 0.0, 0.0)` | `(1.0, 1.0, 1.0)` |
| `--h-couplings` $(h_X, h_Y, h_Z)$ | `(0.0, 0.0, -1.0)` | `(1.0, 1.0, 1.0)` |

Table 3: Additional non-default parameters for reproducing the experiments in Section 4.1.

| Command | Values |
|---|---|
| `--hyperopt` | `optim=grid,steps=80,interval=75*1+100*25,loss=mll` |
| `--kernel-params` | `sigma_0=1.0,gamma=2.0` |

## H  Experimental Details

Every numerical experiment, unless stated otherwise, consists of 50 independent seeded trials. Every seed (trial) starts with one datapoint, $\mathcal{D}^0 = (\hat{\boldsymbol{x}}^0, \hat{y}^0)$, with $\boldsymbol{x}^0$ being an initial point uniformly drawn from $[0, 2\pi)^D$, and $y^0$ being the associated energy at $\boldsymbol{x}^0$ evaluated on the quantum computer. Those 50 initial pairs are cached and, when an optimization trial starts with a given seed, the corresponding cached initial pair is loaded. This allows a fair comparison of different optimization methods: all methods start from the same set of initialization points.

The Qiskit [43] open-source library is used to classically simulate the quantum computer, whereas the rest of the implementation uses pure Python. All numerical experiments have been performed on Intel Xeon Silver 4316 @ 2.30GHz CPUs, and the code with instructions on how to run and reproduce the results is publicly available on GitHub [44].

**VQE kernel analysis (Section 4.1):**  We compare the performance of our proposed VQE-kernel for VQE with the Ising Hamiltonian, i.e., the Heisenberg Hamiltonian (13) with the coupling parameters set to

$$J_X = -1, \ J_Y = 0, \ J_Z = 0, \qquad\qquad h_X = 0, \ h_Y = 0, \ h_Z = -1 \qquad (45)$$

and open boundary conditions (see Table 2). For the variational quantum circuit $G(\boldsymbol{x})$, we use a $(Q = 3)$-qubit, $(L = 3)$-layered `Efficient SU(2)` circuit. In this case, the search domain is $\boldsymbol{x} \in [0, 2\pi)^{24}$, according to Eq. (31). The number of readout shots is set to $N_{\text{shots}} = 1024$. The baseline kernels are the Gaussian-RBF kernel,

$$k^{\text{RBF}}(\boldsymbol{x}, \boldsymbol{x}') = \sigma_0^2 \exp\left(-\frac{\|\boldsymbol{x} - \boldsymbol{x}'\|^2}{2\gamma^2}\right), \qquad (46)$$

and the Periodic kernel [67],

$$k^{\text{period}}(\boldsymbol{x}, \boldsymbol{x}') = \sigma_0^2 \exp\left(-\sum_{d=1}^{D} \frac{1}{2\gamma^2} \sin^2\left(\frac{x_d - x_d'}{2}\right)\right), \qquad (47)$$

which are compared with our VQE kernel (9) in terms of the standard BO performance in Figure 2. Each kernel has two hyperparameters, the prior variance $\sigma_0^2$ and the smoothness parameter $\gamma$. For all three kernels, the prior variance is fixed to $\sigma_0^2 = 1$, and the smoothness parameter $\gamma$ is automatically tuned by marginal likelihood maximization (grid search) in each iteration in the early stage ($t = 0, \ldots, 75$), and after every 100 iterations in the later stage ($t = 76, \ldots$).

For the standard BO, we used the EI acquisition function, which is maximized by L-BFGS [45]. In the code, the SciPy [68] implementation of L-BFGS was used and all experiments were run using the same default parameter set in the code. Detailed commands for reproducing the results can be found in Table 3.

Table 4: Standard choice of EMICoRe hyperparameters for experiments in Section 4.2 and Appendix I (unless specified otherwise).

| | **General params** | |
|---|---|---|
| `--n-qbits` | {3,5,7} | # of qubits |
| `--n-layers` | {3,3,5} | # of circuit layers |
| `--circuit` | esu2 | Circuit name |
| `--pbc` | False | Open Boundary Conditions |
| `--n-readout` ($T_\mathrm{N}$) | {100,300,500} | # iterations for BO |
| `--n-iter` ($T_\mathrm{MI}$) | {100,300,500} | # iterations for BO |
| `--kernel` | vqe | Name of the kernel |
| `--hyperopt` | **Hyperparams optimization** | |
| `optim` | grid | Grid-search optimization of $\gamma$ |
| `max_gamma` | 20 | Max value for $\gamma$ |
| `interval` | 100*1+20*9+10*100 | Scheduling for grid-search |
| `steps` | 120 | # steps in grid |
| `loss` | mll | Loss type |
| `--acq-params` | **EMICoRe params** | |
| `func` | func=ei | Base acq. func. type |
| `optim` | optim=emicore | Optimizer type |
| `pairsize` ($J_\mathrm{SG}$) | 20 | # of candidate points |
| `gridsize` ($J_\mathrm{OG}$) | 100 | # of evaluation points |
| `corethresh` ($\kappa$) | 1.0 | CoRe threshold $\kappa$ |
| `corethresh_width` ($T_\mathrm{Ave}$) | 10 | # averaging steps to update $\kappa$ |
| `coremin_scale` ($C_0$) | 0.0 | coefficient $C_0$ in eq. (35) |
| `corethresh_scale` ($C_1$) | 1.0 | coefficient $C_1$ in eq. (35) |
| `samplesize` ($N_\mathrm{MC}$) | 100 | # of MC samples |
| `smo-steps` ($T_\mathrm{NFT}$) | 0 | # of initial NFT steps |
| `smo-axis` | True | Sequential direction choice |

**EMICoRe analysis (Section 4.2):** In this experiment, we compare the optimization performance of our *NFT-with-EMICoRe* with the NFT baselines (sequential and random) on VQE with both the Ising Hamiltonian, for which the coupling parameters are given in Eq. (45), and the Heisenberg Hamiltonian, for which the coupling parameters are set to

$$J_X = 1, \, J_Y = 1, \, J_Z = 1 \qquad\qquad h_X = 1, \, h_Y = 1, \, h_Z = 1 \,. \qquad (48)$$

For the variational quantum circuit $G(\boldsymbol{x})$, we use a ($Q = 5$)-qubit, ($L = 4$)-layered `Efficient SU(2)` circuit with open boundary conditions, giving ($D = 40$)-dimensional search domain. The number of readout shots is set to $N_\mathrm{shots} = 1024$ in the experiment shown in Figure 3. In the same format as Figure 3, Figures 9–17 in Appendix I.3 compare EMICoRe with the baselines for different setups of $Q$, $L$, and $N_\mathrm{shots}$.

For NFT-with-EMICoRe (Algorithm 2 and Algorithm 3), where the VQE kernel is used, the prior standard deviation is set to a value roughly proportional to the number of qubits $Q$; specifically for $Q = 5$ we set $\sigma_0 = 6$. The smoothness parameter $\gamma$ is automatically tuned by marginal likelihood maximization in each iteration in the early stage ($t = 0, \ldots, 100$), after every 9 iterations in the middle phase ($t = 101, \ldots, 280$), and after every 100 iterations in the last phase ($t = 281, \ldots$). The other parameters are set to $T_\mathrm{MI} = 300$, $J_\mathrm{SG} = 20$, $J_\mathrm{OG} = 100$, $N_\mathrm{MC} = 100$, $T_\mathrm{NFT} = 0$ and $T_\mathrm{Ave} = 10$. All relevant hyperparameters are collected in Table 4 along with the corresponding flags in our code.

The command options that specify the VQE setting, the kernel optimization schedule, and the other parameter settings for EMICoRe, are summarized in Table 2 and Table 4.

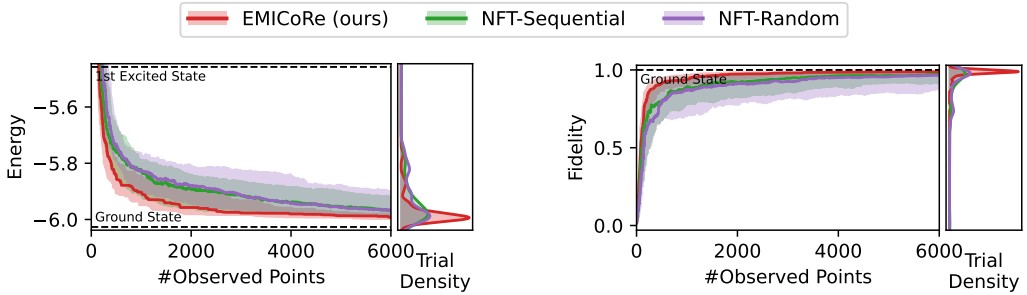

Figure 7: Energy (left) and fidelity (right) for EMICoRe (ours) and the baselines, NFT-sequential and NFT-random, up to 6000 observations. Results are for the Ising Hamiltonian with a $(L = 3)$-layered $(Q = 5)$-qubits quantum circuit and $N_{\text{shots}} = 1024$.

Table 5: Energy and fidelity achieved after 6000 observations in the experiment in Figure 7. The best results are highlighted in bold, while $\downarrow$ ($\uparrow$) indicates lower (higher) values are better.

| Algorithm | Energy $\downarrow$ | Fidelity $\uparrow$ |
|---|---|---|
| **EMICoRe (ours)** | $\mathbf{-5.97 \pm 0.05}$ | $\mathbf{0.98 \pm 0.04}$ |
| NFT-random | $-5.92 \pm 0.08$ | $0.92 \pm 0.09$ |
| NFT-sequential | $-5.93 \pm 0.09$ | $0.92 \pm 0.16$ |

# I  Additional Experiments

Here, we report on additional experimental results.

## I.1  Convergence to Ground State

The experiments from Section 4.2, compared the performance between our EMICoRe and the NFT baselines up to 600 observed points, where the optimization has not yet converged. Here, we perform a longer optimization with up to 6000 observations to confirm the ability of EMICoRe to converge to the ground state. In Figure 7 the energy and fidelity plots show the optimization progress for the Ising model with a $(L = 3)$-layered $(Q = 5)$-qubits quantum circuit and $N_{\text{shots}} = 1024$ readout shots. The portrait plot on the right shows the distribution over 50 independent trials of the final solutions after 6000 observations. The mean and the standard deviation of the achieved energy and fidelity are summarized in Table 5. We observe that EMICoRe achieves an average fidelity above $95\%$ after 1000 observations, and reaches $98\%$ fidelity at 4000 observations. In contrast, the NFT baselines require all 6000 observations in order to achieve a fidelity of $92\%$, exhibiting a much slower convergence. This result confirms that EMICoRe robustly converges to the ground state, independent of the individual trial's initialization.

Note that the GP regression exhibits cubic computational complexity with respect to the number of samples, thus significantly slowing down the optimization process with thousands of observed points. As a remedy, we limit the number of utilized samples by discarding old observations, i.e., we choose *inducing points* for the GP based on the chronological order of observations. We found in preliminary experiments that choosing the last 100 observations as inducing points is sufficient to achieve good results. In the experiments above (see Figure 7), we keep the number of inducing points above 100 and below 120, where we discard the 20 oldest points when exceeding a number of 120 observations. This strategy is implemented in our public code [44] through the option `--inducer last_slack:retain=100:slack=20`, where `last_slack` indicates the criterion to choose the inducing points, and where `retain` and `slack` can be used to specify the minimum number of points retained and the number of samples that can be observed beyond the minimum, before discarding. Hence, the model discards `slack=20` observations when the number of observed points equals to the sum of the two, i.e., `slack + retained`.

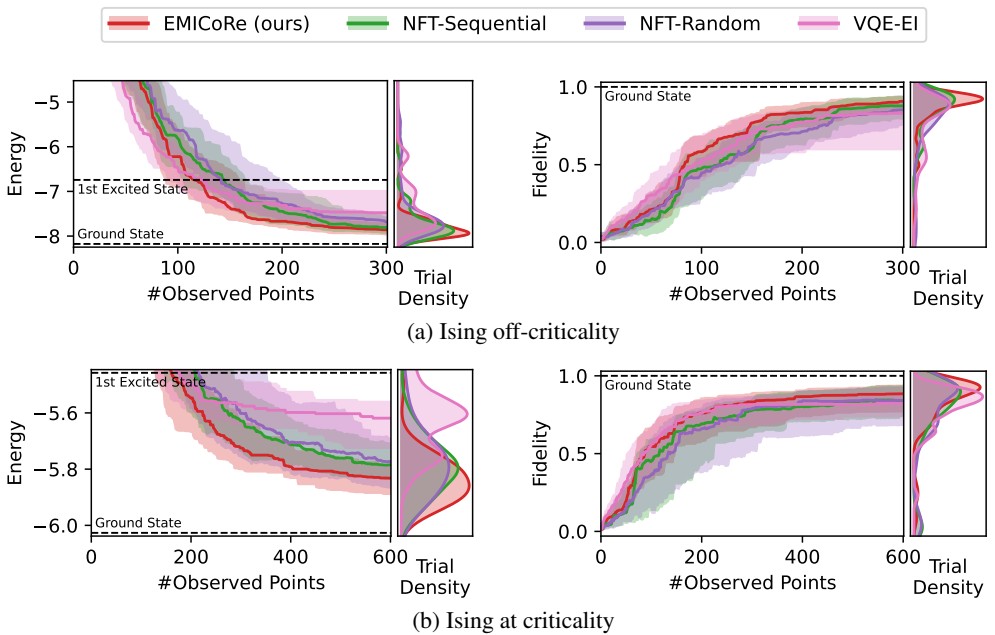

Figure 8: Energy (left) and fidelity (right) for EMICoRe (ours) compared to three different baselines: NFT-sequential, NFT-random, and EI with VQE kernel. Results for the Ising Hamiltonian off-criticality and at criticality are shown in the top and bottom rows respectively.

## I.2 Ising Hamiltonian Off-Criticality

In Section 4, we focused on the Ising and Heisenberg Hamiltonians with the parameters, $J = (J_X, J_Y, J_Z)$ and $h = (h_X, h_Y, h_Z)$ in Equation (13), set to criticality in the thermodynamic limit. Such choices are expected to be most challenging for the VQE optimization because the corresponding ground states tend to be highly entangled due to the quantum phase transition. As an ablation study, we here conduct experiments for an off-critical setting. Specifically, we evaluate the optimization performance for the Ising Hamiltonian *off-criticality* $\{J = (0, 0, -1); h = (1.5, 0, 0)\}$. Figure 8 (top) shows the energy (left) and the fidelity (right) achieved by EMICoRe (ours), NFT-sequential, NFT-random, and EI with VQE kernel after 600 iterations for a $(L = 3)$-layered $(Q = 5)$-qubits quantum circuit with $N_{\text{shots}} = 1024$ readout shots. For comparison, we also show in Figure 8 (bottom) the performance for the Ising Hamiltonian at criticality $\{J = (-1, 0, 0); h = (0, 0, -1)\}$. We observe that the off-criticality setting is significantly easier, while the plain BO with EI (without EMICoRe) falls somewhat short, although closely behind NFT-random at 600 observed points.

## I.3 Different Setups for Qubits and Layers

Here, we compare the performance of the baselines (NFT-sequential and NFT-random) to our NFT-with-EMICoRe under different settings of $Q$, $L$, and $N_{\text{shots}}$. All figures in this appendix (Figures 9–17) are shown in the same format as Figure 3: for each figure, the energy (left column) and the fidelity (right column) are shown for the Ising (top row) and the Heisenberg (bottom row) Hamiltonians. In each panel, the left plot shows the optimization progress with the median (solid) and the 25- and 75-th percentiles (shadow) over the 50 seeded trials, as described in Appendix H, as a function of the observation costs, i.e., the number of observed points. The portrait square on the right shows the distribution of the final solution after 200, 600, and 1000 observations have been performed, respectively, for $Q = 3$, 5, and 7 qubit cases. As mentioned earlier, the prior standard deviation $\sigma_0$ is set roughly proportional to $Q$. Specifically we use $\sigma_0 = 4, 6, 9$ for $Q = 3, 5, 7$, respectively.

**3-qubit setup:** Figures 9–11 show results for $Q = 3$, $L = 3$, and $N_{\text{shots}} = 256, 512, 1024$. Given the relatively low-dimensional problem with only $D = 24$, the convergence rate is comparable for both baselines and our approach. However, the red sharp peak in the density plot of the final solutions

(right portrait square) in each panel implies that the robustness against initialization is improved by our NFT-with-EMICoRe approach, thus highlighting its enhanced stability and noise-resiliency.

**5-qubit setup:** Figures 12–14 show results for $Q = 5$, $L = 3$, and $N_{\text{shots}} = 256, 512, 1024$. The case for $N_{\text{shots}} = 1024$ is identical to Figure 3 in the main text. For all noise levels ($N_{\text{shots}} = 256, 512, 1024$), our EMICoRe (red) consistently achieves lower energy and higher fidelity compared to the baselines, thus demonstrating the superiority of our NFT-with-EMICoRe over NFT [11]. Remarkably, we observe that in high-noise-level cases, such as for the Heisenberg Hamiltonian for $N_{\text{shots}} = 256, 512$ (bottom rows of Figure 12 and Figure 13), the achieved energy by the state-of-the-art baselines (purple and green) fail to even surpass the energy level of the first excited state for the 600 observed data points, whereas EMICoRe successfully accomplishes this task.

**7-qubit setup:** Figures 15–17 present results for $Q = 7$, $L = 5$, and $N_{\text{shots}} = 256, 512, 1024$. Again, NFT-with-EMICoRe consistently outperforms the baselines in *all* experimental setups. Given the increased complexity associated with $Q = 7$ and $D = 84$, the optimization process becomes more challenging, necessitating a greater number of observed points to approach the ground-state energy. Nonetheless, even with just 1000 observed points, NFT-with-EMICoRe already exhibits significant superiority over NFT [11], particularly in high-noise scenarios such as $N_{\text{shots}} = 256$. We also observed that the optimization process faces difficulties with the Heisenberg Hamiltonian case. We attribute this behavior to the greater complexity of the latter task and defer further analysis of this regime to future studies.

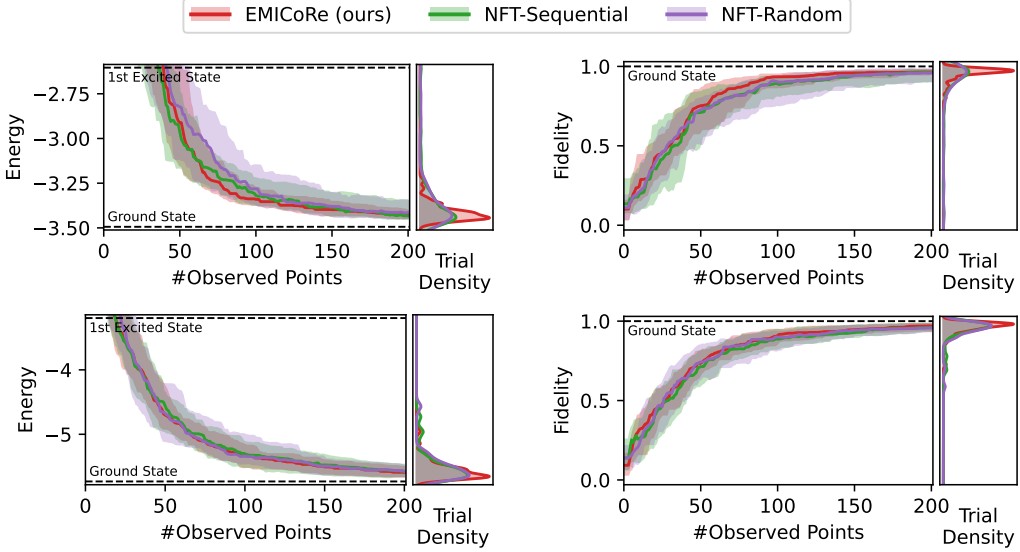

Figure 9: Comparison (in the same format as Figure 3) between our NFT-with-EMICoRe (red) and the NFT baselines (green and purple) in VQE for the Ising (top row) and Heisenberg (bottom row) Hamiltonians with the ($L = 3$)-layered ($Q = 3$)-qubit quantum circuit (thus, $D = 24$) and $N_{\text{shots}} = 256$.

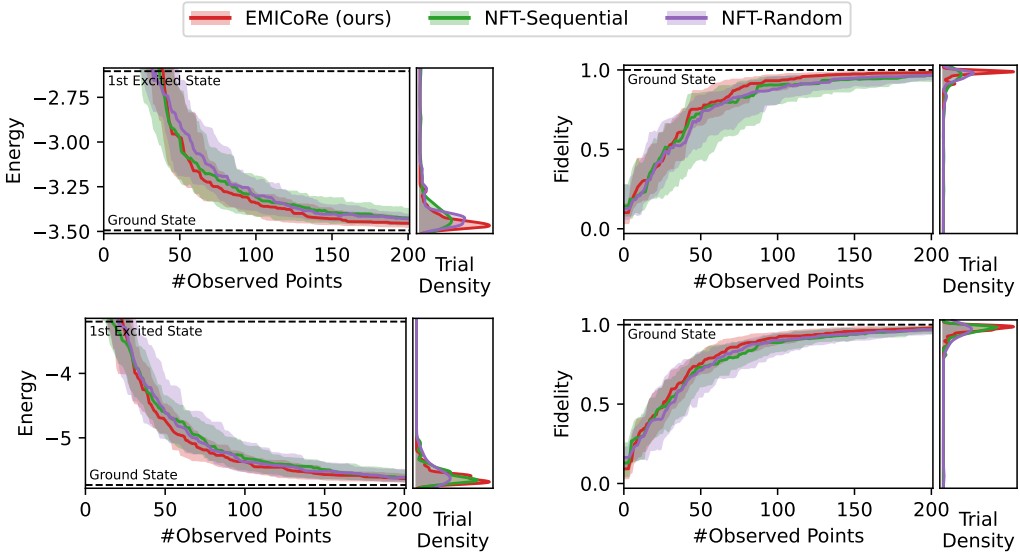

Figure 10: Same comparison as in Fig. 9, with the $(L = 3)$-layered $(Q = 3)$-qubit quantum circuit (thus, $D = 24$) and $N_{\text{shots}} = 512$. The NFT-with-EMICoRe (red) and NFT baselines (green and purple) are shown for both Ising (top row) and Heisenberg (bottom row) Hamiltonians.

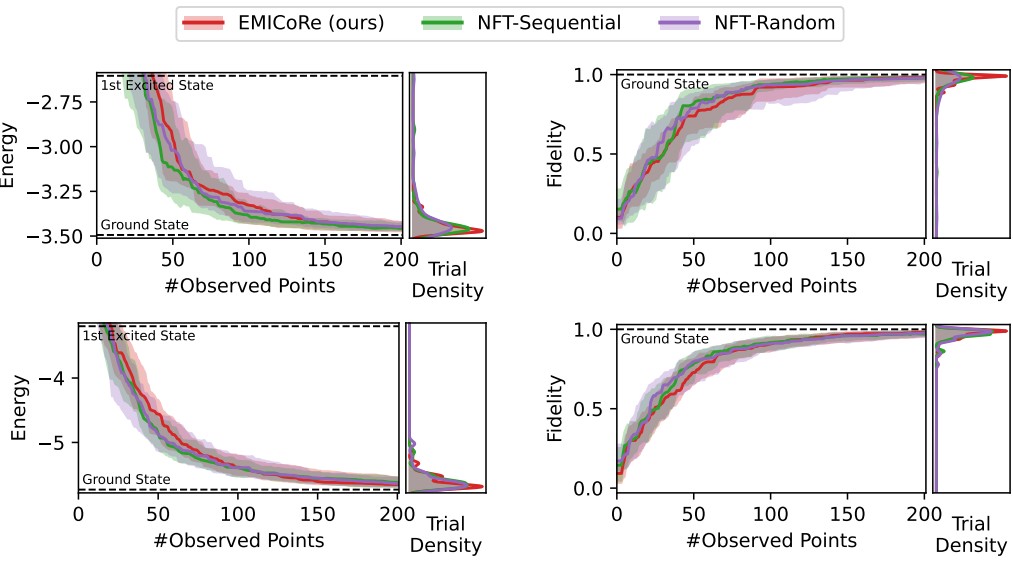

Figure 11: Same comparison as in Fig. 9, with the $(L = 3)$-layered $(Q = 3)$-qubit quantum circuit (thus, $D = 24$) and $N_{\text{shots}} = 1024$. The NFT-with-EMICoRe (red) and NFT baselines (green and purple) are shown for both Ising (top row) and Heisenberg (bottom row) Hamiltonians.

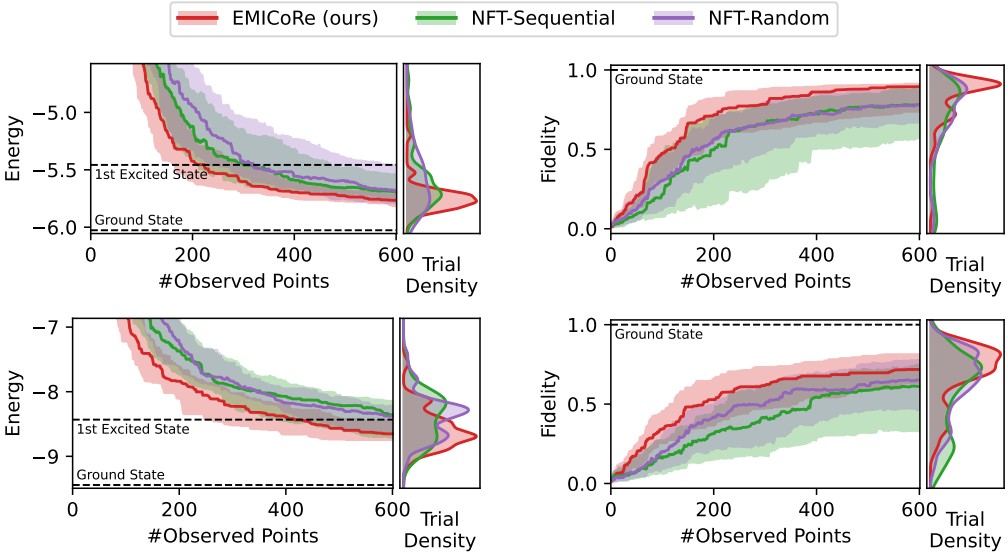

Figure 12: Same comparison as in Fig. 9, with the $(L = 3)$-layered $(Q = 5)$-qubit quantum circuit (thus, $D = 40$) and $N_{\text{shots}} = 256$. The NFT-with-EMICoRe (red) and NFT baselines (green and purple) are shown for both Ising (top row) and Heisenberg (bottom row) Hamiltonians.

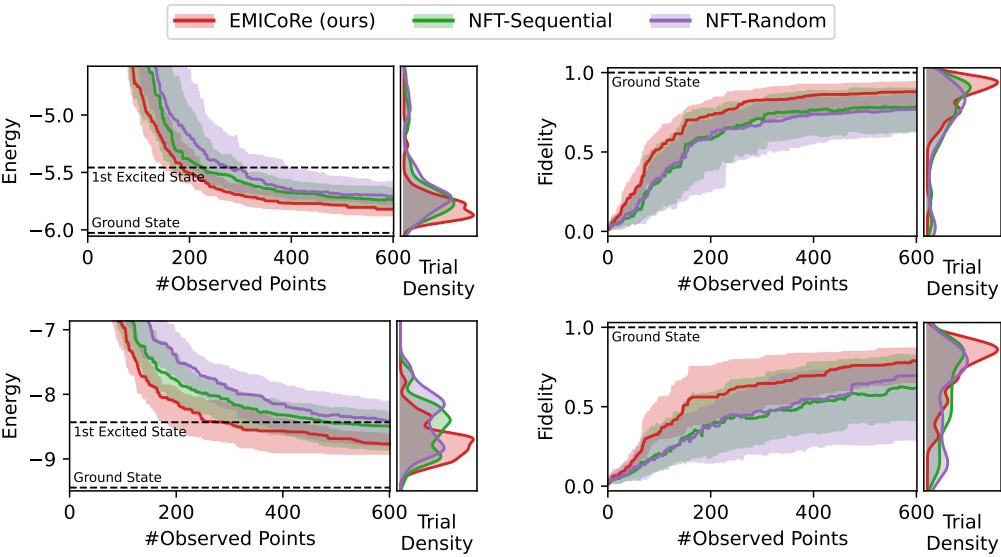

Figure 13: Same comparison as in Fig. 9, with the $(L = 3)$-layered $(Q = 5)$-qubit quantum circuit (thus, $D = 40$) and $N_{\text{shots}} = 512$. The NFT-with-EMICoRe (red) and NFT baselines (green and purple) are shown for both Ising (top row) and Heisenberg (bottom row) Hamiltonians.

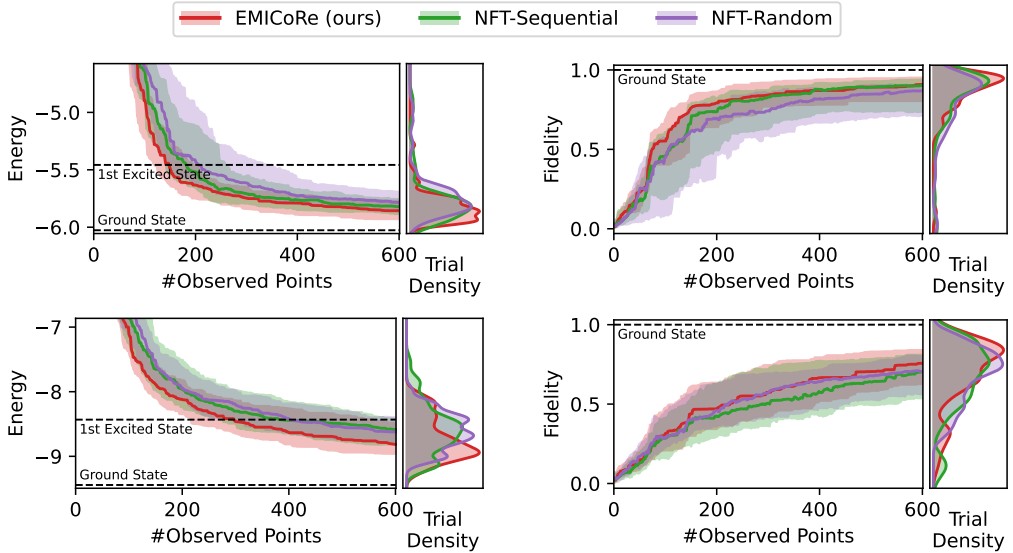

Figure 14: Same comparison as in Fig. 9, with the $(L = 3)$-layered $(Q = 5)$-qubit quantum circuit (thus, $D = 40$) and $N_{\text{shots}} = 1024$. The NFT-with-EMICoRe (red) and NFT baselines (green and purple) are shown for both Ising (top row) and Heisenberg (bottom row) Hamiltonians.

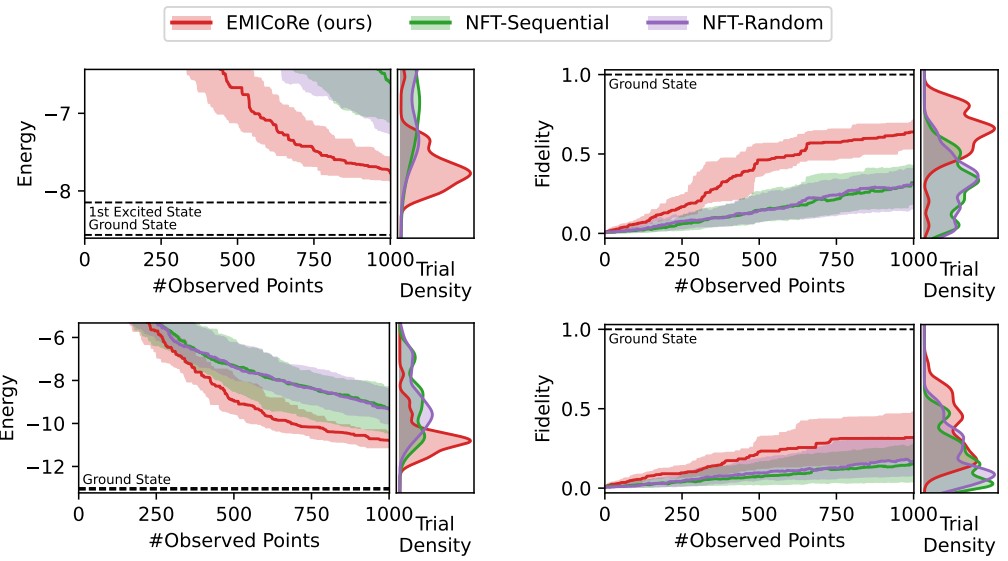

Figure 15: Same comparison as in Fig. 9, with the $(L = 5)$-layered $(Q = 7)$-qubit quantum circuit (thus, $D = 84$) and $N_{\text{shots}} = 256$. The NFT-with-EMICoRe (red) and NFT baselines (green and purple) are shown for both Ising (top row) and Heisenberg (bottom row) Hamiltonians.

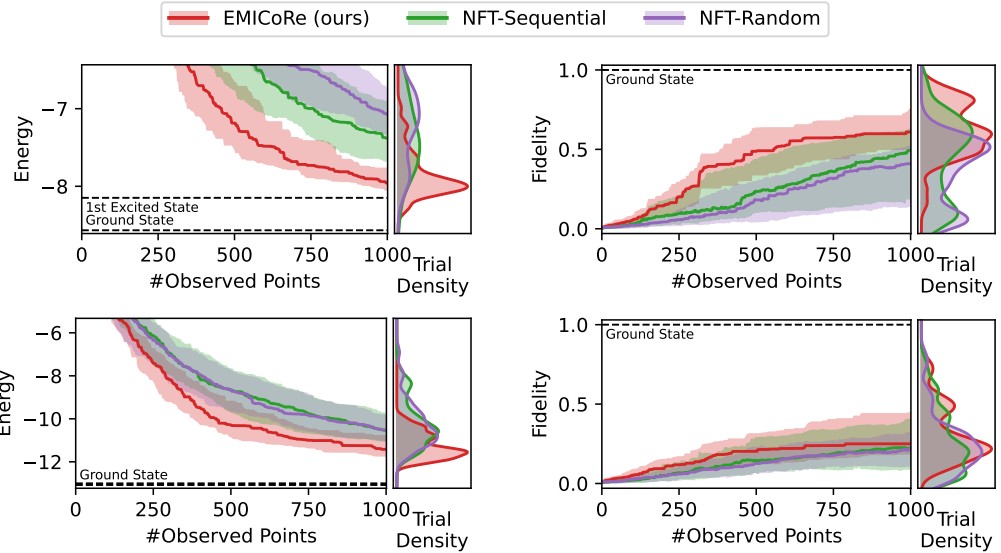

Figure 16: Same comparison as in Fig. 9, with the $(L = 5)$-layered $(Q = 7)$-qubit quantum circuit (thus, $D = 84$) and $N_{\text{shots}} = 512$. The NFT-with-EMICoRe (red) and NFT baselines (green and purple) are shown for both Ising (top row) and Heisenberg (bottom row) Hamiltonians.

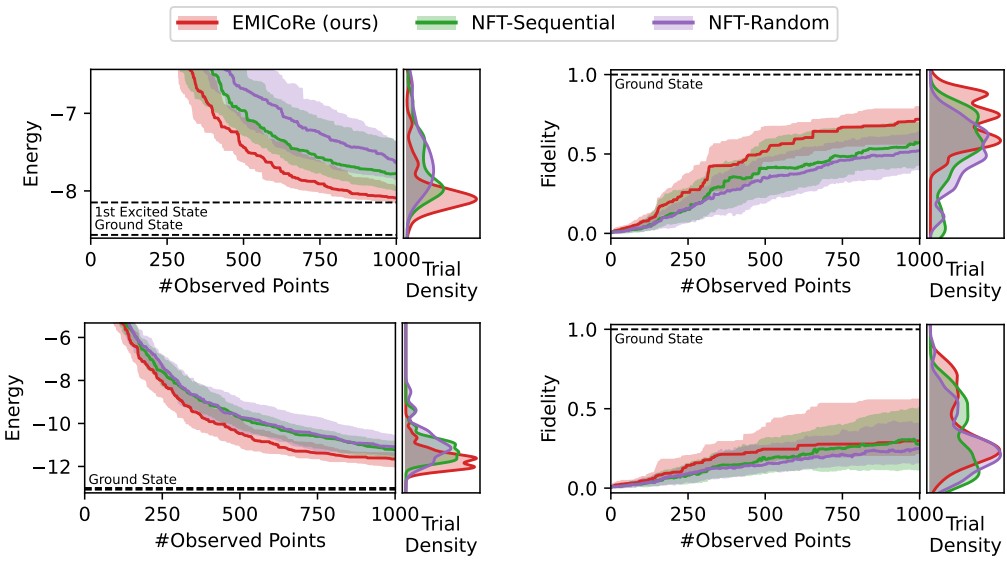

Figure 17: Same comparison as in Fig. 9, with the $(L = 5)$-layered $(Q = 7)$-qubit quantum circuit (thus, $D = 84$) and $N_{\text{shots}} = 1024$. The NFT-with-EMICoRe (red) and NFT baselines (green and purple) are shown for both Ising (top row) and Heisenberg (bottom row) Hamiltonians.

