^{-1}\cos x_d \\ \vdots \\ \gamma^{-V_d}\cos(V_d x_d) \\ \gamma^{-1}\sin x_d \\ \vdots \\ \gamma^{-V_d}\sin(V_d x_d) \end{pmatrix} \right). \tag{34}$$

*Proof.* Similarly to the *exclusive parameterization* case, we have

$$k^{gVQE}(\boldsymbol{x}, \boldsymbol{x}') = \sigma_0^2 \prod_{d=1}^{D} \left( \frac{1 + \sum_{v=1}^{V_d} \gamma^{-2v} \cos\left(v(x_d - x_d')\right)}{1 + \sum_{v=1}^{V_d} \gamma^{-2v}} \right)$$

$$= \sigma_0^2 \left( 1 + \sum_{v=1}^{V_d} \gamma^{-2v} \right)^{-D} \prod_{d=1}^{D} \left( 1 + \sum_{v=1}^{V_d} \gamma^{-2v} \