# OpenReview forum: "Physics-Informed Bayesian Optimization of Variational Quantum Circuits"
_NeurIPS.cc/2023/Conference — NeurIPS 2023 poster_

### Official Review · Reviewer_nt6b · 2023-06-28

**Soundness:** 1 poor
**Presentation:** 3 good
**Contribution:** 2 fair
**Rating:** 6
**Confidence:** 4

**Summary:**

The paper introduces a new approach for BO of VQEs. BO can be a good match for this problem since it models the noise (measurement and circuit level) of quantum circuits. The main idea of the paper is to use a kernel that is adapted to the form of the VQE objective assumed when variational parameters are associated to single qubit gates. The paper also introduces EMICoRE and NFT-with-EMICoRE, acquisition function and parameter update strategies. Overall, the idea of using an adapted kernel for this task is well motivated, the paper is well written and presents numerical evaluation of their method. I think however that in the current state, the experimental evaluation is too limited to be convincing of the promise of this method with respect to other optimisation algorithms for VQEs.

**Strengths:**

The paper addresses an important problem in quantum computing. It is well written and accessible for both quantum computing and ML communities I think. It introduces the following technical innovations:
- Kernel adapted to VQEs: the kernel decomposition based on the form of the cost function of VQEs is a sound contribution.
- NFT-with-EMICoRE: a new BO algorithm that builds on NFT and a new acquisition function.



**Weaknesses:**

1. Specific ansatz: the paper deals only with parametrised single qubit gates. While I believe this is not a theoretical limitation - as one can always decompose any unitary as a product of single qubit gates and CNOTs - it is not clear to me how practical is this parametrisation. The authors could add an explanation of whether this is a limitation for practitioner and what steps one needs to do to apply their algorithm to a generic parametrised quantum circuit
2. Limited experimental evaluation: the paper only discusses Q=3 and 5 qubit systems, while classically one could simulate easily bigger systems. Also the paper does not compare against other methods in the literature, it only compare against NFT. More benchmarking for larger systems and against other techniques would be required to assess the promise of the method.
3. Unclear benefit of the kernel: looking at figure 2, it is unclear that the red curve is better than blue curve - in fact on the left figure it looks to me that RBF is doing better.
4. Missing ablation for the EMICoRE acquisition function. One contribution of the paper is to introduce this new acquisition function, but ablations showing its importance are missing.

Minor:
- $\mu_X$ depends on y as well in section 2.1
- "the the" in line 308
- unclear what conjugacy means in line 97

**Questions:**

1. Could you benchmark your method against other methods for VQE optimisation than NFT? And for larger systems?
2. Why is the ground state energy not reached in figure 3 say? I would expect that it is possible to find a VQE ansatz that can solve this easy problem.
3. Can you add ablations for the EMICoRE acquisition function? Have you tried different heuristics for updating $\kappa$?
4. Can you explain how your method can be used to optimise general parametrised circuits beyond those that have single qubit gates?


**Limitations:**

- limited experiments and benchmarking
- limited explanation of their choice of parametrised circuits

---

> ### Author Rebuttal · Authors · 2023-08-07
>
> We thank the referee for highlighting the strengths of our work. We hereby address the concerns raised in the review:
>
> ## **Weaknesses**
> > 1. Specific ansatz: ...
>
> The set of single-qubit rotation gates and entangling CNOT gates are indeed universal, and allow for synthesizing all unitary operations on a subset of qubits in an efficient manner, see [1]. More generally, this is the essence of the Solovay-Kitaev theorem [2]. Moreover, most commercially available superconducting quantum devices offer single-qubit rotation gates and CNOT entangling gates, such that our ansatz does not pose a restriction in practice. In fact, Qiskit, one of the most prevalent SDKs for developing quantum programs, offers the ansatz structure we use as *"a heuristic pattern that can be used to prepare trial wave functions for variational quantum algorithms or classification circuit for machine learning"* (see  [qiskit.org/documentation](https://qiskit.org/documentation/stubs/qiskit.circuit.library.EfficientSU2.html)). Furthermore, our work is not limited to the type of ansätze we studied in the manuscript. The only restriction is that the parametric gates in the ansatz are of the form $R(\theta) = \exp(-i\theta G)$, where $G$ is an arbitrary gate operation that fulfills $G^2 = \mathbf{1}$, see Eq. (2) in the NFT paper [3]. This comprises any gate where $G$ is a tensor product of Pauli operators, for example parametric entangling gates of the form $\exp(-i\theta Z_l\otimes Z_k)$ with $Z_j$ the Pauli $Z$-matrix acting on qubit $j$.
>
> > 2. Limited experimental evaluation: ...
>
> In the paper's Appendix G, we show results for Q=7 in Figs. 13-15. The method we compare to, NFT [3], is widely accepted as the current state-of-the-art. Specifically, in [3], the authors showed superior performance compared to many gradient-based and gradient-free methods, i.e., SPSA[4], BFGS[5], N-M[6], Powell[7], CG[8]. We, therefore, refrained from including these inferior baselines to streamline the presentation. Our numerical experiments clearly establish that EMICoRe gains favorable scaling with the system size, i.e., our method can beat the state-of-the-art method by an increasing margin as the number of qubits is increased, see Figs. 7-14 in Appendix G.
>
> > 3. Unclear benefit of the kernel: ...
>
> In Table R3 below, we show that our VQE kernel reaches, on average, lower energy and higher fidelity, with smaller variance, when compared to RBF and periodic kernels. Furthermore, the VQE kernel is a necessary ingredient in order to perform the EMICoRe strategy which is proven to outperform the state-of-the-art baseline in our experiments.
>
> > 4. Missing ablation ...
>
> In Fig. 2 in the main paper, we evaluated the performance gain of our VQE kernel without EMICoRe.
> To directly show the ablation we conducted additional experiments with $Q=5$ and $L=3$ in the rebuttal PDF. We showcase the benefits of EMICoRe by comparing it to the widely adopted Expected Improvement (EI) acquisition function for two types of systems, i.e., the Ising model at and away from criticality.
>
> ## **Questions**
> > 1. Benchmark, larger systems
>
> It is well-known that NFT is the current state-of-the-art algorthm for optimizing VQEs. We therefore refrained from including other inferior baselines.
> Furthermore, as mentioned above, our method can beat the state-of-the-art method by an increasing margin as the system size increases. We refer to our reply in the **Weakness** section above.
>
> > 2. Ground state energy in figure 3
>
> In Figure 3 from the main paper, our main goal was to demonstrate the faster optimization of EMICoRE compared to NFT at a fixed quantum measurement budget. In Fig. R2 in the attached PDF, we demonstrate that our algorithm can reach the ground state for a larger measurement budget. We will include these experiments in the revised manuscript.
>
> > 3. Ablations
>
> We performed additional experiments in the direction of an ablation study for EMICoRe. This includes comparing to other acquisition functions, e.g., EI, as shown in Fig. R1 in the rebuttal PDF, and performing an in-depth analysis for setting the CoRe threshold $\kappa$, shown in the Table R1 from the general rebuttal above.
>
> > 4. General parametrised circuits
>
> As we pointed out in the **Weakness** section, our method is not limited to the parametric gates we use in our ansatz nor is it limited to single-qubit parametric gates. In general, our approach can be applied to any parametric gate of the form $R(\theta) = \exp(-i\theta G)$ with $G$ a gate that has the property $G^2=\mathbf{1}$, see Eq. (2) in the NFT paper [3]. This includes a wide range of gates, in particular, $G$ can be an arbitrary Pauli string $\otimes_{q=1}^QP_q$, where $P_q$ is one of the Pauli matrices {$\mathbf{1}, X, Y, Z$} acting on qubit $q$. More specifically, this set also includes the entangling gates $R_{XX}=\exp(-i\theta X_l\otimes X_k )$ and $R_{ZZ}=\exp(-i\theta Z_l\otimes Z_k )$, which are more commonly realized in trapped ion setups.
>
>
> ### Table R3: Energy and Fidelity for 600 observed points and different kernels, as shown in Fig. 2. Best results highlighted in bold. For energy lower is better. For fidelity, higher is better.
> | Kernel | Energy | Fidelity |
> |-|-|-|
> | **vqe (ours)** | **-3.306890 ± 0.117974** | **0.926119 ± 0.047960** |
> | rbf | -3.269373 ± 0.145235 |  0.895492 ± 0.084795 |
> | periodic | -3.232246 ± 0.282348 | 0.901823 ± 0.095861 |
>
> ## **References**
> - [1] A. Barenco A. et al., Phys. Rev. A 52, 3457 (1995).
> - [2] A. Y. Kitaev,  Russ. Math. Surv. 52 _1191_ (1997).
> - [3] K. Nakanishi et al., Phys. Rev. Research 2, 043158 (2020).
> - [4] J. C. Spall, IEEE Transactions on Automatic Control, 37(3):332–341, 1992.
> - [5] D. C. Liu, J. Nocedal, Mathematical Programming volume 45, pages 503–528 (1989).
> - [6] J. A. Nelder and R. Mead, The Computer Journal 7, 308 (1965).
> - [7] M. J. D. Powell, The Computer Journal, Volume 7, Issue 2, 1964, Pages 155–162 (1964).
> - [8] R. Fletcher and C. M. Reeves, Computer Journal 7 (1964).

---

> > ### Comment · Reviewer_nt6b · 2023-08-17
> >
> > Thank you, the rebuttal addressed my concerns and I increased the score.

---

### Official Review · Reviewer_1i6G · 2023-07-01

**Soundness:** 2 fair
**Presentation:** 2 fair
**Contribution:** 3 good
**Rating:** 5
**Confidence:** 4

**Summary:**

In this work, the authors integrate a quantum kernel method with the EMICoRe architecture to further improve the NFT framework of Bayesian Optimization. The simulation results show the advantages of the proposed approach.

**Strengths:**

(1) The method of leveraging the quantum method for Bayesian Optimization is interesting.

(2) The investigation of how to incorporate the quantum kernel method is significant.



**Weaknesses:**

(1) Since no parametric circuits are implemented in the quantum kernel model shown in Eq. (9), the proposed VQE-kernel is nothing but a quantum kernel learning method, which has been comprehensively studied in previous work in Refs. [1-3]. In particular, Ref. [4] has exhibited the use of quantum kernel learning for improving the performance of Bayesian optimization. --- resolved

(2) The circuit diagram for the quantum kernel learning is not provided such that the experiments cannot be easily reproduced. --- resolved

(3) There are two main contributions to this work: the use of quantum kernel learning and an introduction to EMICoRe. Although the simulation results demonstrate the performance improvement, it is still unknown where the performance gains come from, and the quantum advantages of the quantum kernel are not analyzed at all. --- resolved

[1] Havlíček, Vojtěch, et al. "Supervised learning with quantum-enhanced feature spaces." Nature 567.7747 (2019): 209-212

[2] Wang, Xinbiao, et al. "Towards understanding the power of quantum kernels in the NISQ era." Quantum 5 (2021): 531

[3] Blank, Carsten, et al. "Quantum classifier with tailored quantum kernel." npj Quantum Information 6.1 (2020): 41

[4] Rath, Yannic, Aldo Glielmo, and George H. Booth. "A Bayesian inference framework for compression and prediction of quantum states." The Journal of Chemical Physics 153.12 (2020)

**Questions:**

(1) What are the quantum advantages of the quantum kernel method? Since the quantum kernel method is combined with another proposed method `EMICoRe', we do not know which part contributes to the improvement of the Bayesian Optimization method. --- resolved

**Limitations:**

It would be better to reformulate the quantum approach in the authors' proposed Bayesian Optimization framework.

---

> ### Author Rebuttal · Authors · 2023-08-07
>
> We thank the reviewer for the feedback. However, it seems as if the referee misunderstood the paper, as our work cannot be characterized as quantum kernel learning. In essence, quantum kernel leverages a quantum circuit to calculate a certain kernel, i.e., the calculation of the kernel is non-classical. In our work, we use classical kernels. Specifically, we propose a novel *classical* kernel which is specifically tailored to the VQE setup. More precisely, the corresponding feature vectors precisely correspond to the energy landscape modeled by the VQE. This is an important and novel contribution in its own right. We go beyond this insight and deploy this kernel in Bayesian optimization. Specifically, we propose EMICoRe and demonstrate that it leads to superior performance when compared to the current state-of-the-art method for optimizing VQE, i.e. NFT. In the following, we reply in more detail to the reviewer's comments:
>
> ## **Weaknesses**
>
> > Since no parametric circuits are implemented in the quantum kernel model shown in Eq. (9), the proposed VQE-kernel is nothing but a quantum kernel learning method, which has been comprehensively studied in previous work in Refs. [1-3]. In particular, Ref. [4] has exhibited the use of quantum kernel learning for improving the performance of Bayesian optimization.
>
> As discussed above, our work cannot be characterized as quantum kernel learning, which is the topic of the suggested references. We have added a discussion of the notable differences with respect to quantum kernel learning to the related work section in the updated version of the manuscript.
>
> > The circuit diagram for the quantum kernel learning is not provided such that the experiments cannot be easily reproduced.
>
> As we do not use a quantum kernel, we cannot provide a circuit diagram for it. In more detail, for quantum kernel learning, one would need two circuits, see Figure 1 of Ref. [2] that the referee has mentioned. In our case, we have a classical kernel. Therefore, we only require one circuit: the Efficient SU(2) circuit, as shown in Figure 6 in the Appendix of our paper for reproducibility. We also would like to emphasize that we provided the code for reproducing all our experiments, including the implementation of the quantum circuit.
>
> > There are two main contributions to this work: the use of quantum kernel learning and an introduction to EMICoRe. Although the simulation results demonstrate the performance improvement, it is still unknown where the performance gains come from, and the quantum advantages of the quantum kernel are not analyzed at all.
>
> As discussed above, our work does not study quantum kernels, thus, no possible quantum advantage from a quantum kernel can be analyzed. Regarding the quantum advantage of VQE and the performance gain, please refer to the following answer.
>
> ## **Questions**
> > What are the quantum advantages of the quantum kernel method? Since the quantum kernel method is combined with another proposed method `EMICoRe`, we do not know which part contributes to the improvement of the Bayesian Optimization method.
>
> As we pointed out in the **Weakness** section, our work does not consider quantum kernels, so no quantum advantage of a quantum kernel can be studied. Instead, our work proposes a *classical* kernel that is uniquely suited for the VQE setup. Specifically, the corresponding feature vectors precisely align with the energy landscape modeled by the VQE. In particular, it retains the property of VQEs that the energy can be determined along an entire line by only fixing three points. This powerful inductive bias allows us to propose EMICoRe, which can outperform the current state-of-the-art method [1]. For the VQE algorithm, it is very well known that the quantum advantage comes from the efficient implementation of the exponentially large Hilbert space using the qubits of a quantum computer. A quantum computer only needs $Q$ qubits to prepare the quantum state $|\psi(\theta)\rangle$ in the VQE algorithm, while a classical computer would require exponentially large resources, $2^Q$, for the same operation.
>
> ## **References**
> - [1] [Nakanishi K. et al., Phys. Rev. Research 2, 043158 (2020).](https://journals.aps.org/prresearch/abstract/10.1103/PhysRevResearch.2.043158)
> - [2] [Wang, Xinbiao, et al. "Towards understanding the power of quantum kernels in the NISQ era." Quantum 5 (2021): 531](https://quantum-journal.org/papers/q-2021-08-30-531/)

---

> > ### Comment · Reviewer_1i6G · 2023-08-17
> > **Follow-up for the rebuttal letter**
> >
> > Thank the authors for providing the rebuttal letter, which helps the reviewer to better understand the paper. Since the authors have resolved all of my major concerns, I have increased my suggested score for the paper.

---

> > > ### Author Response · Authors · 2023-08-20
> > > **Strengths after misunderstanding has been resolved?**
> > >
> > > Dear Reviewer 1i6G,
> > >
> > > Thank you very much for responding to our rebuttal and updating your review.  We are happy to know from your updated review that all the weaknesses and the questions in the original review have been resolved.  We also appreciate that the underlying misunderstanding, i.e., misclassification of our work as quantum kernel learning, has also been resolved.
> > >
> > > However, we wonder why you recommend borderline accept, which only means that "reasons for accept outweigh reasons to reject", although no weakness remains unsolved. Would you mean that the strengths of our paper are insignificant?  This point is unclear to us because the strengths in your review are unchanged from the original review, which are based on a fundamental misunderstanding of our paper, and therefore did not evaluate our contributions (our paper is NOT about (1) quantum method for Bayesian optimization NOR about (2) quantum kernel method BUT about Bayesian optimization for hybrid quantum-classical computing with a novel classical kernel and a novel acquisition function).
> > >
> > > We would appreciate it if you would update the strengths, and clarify why the strengths are not sufficient for acceptance even after all weaknesses have been resolved.
> > >
> > > Sincerely,
> > > Authors

---

### Official Review · Reviewer_WANE · 2023-07-13

**Soundness:** 4 excellent
**Presentation:** 4 excellent
**Contribution:** 3 good
**Rating:** 7
**Confidence:** 3

**Summary:**

The authors propose a method for Bayesian Optimization for Variational Quantum Eigensolvers, which they call NFT with EMICoRe.  This method uses a novel VQE kernel, which constrains the function space of the Gaussian Process underlying the BO to include only valid VQE objective functions (using the representation in Prop. 2, derived from NFT).  The authors also propose a novel acquisition function for the EMICoRe method (Eq. 11), which optimizes over the expected maximum improvement over confident regions.  In their experimentation, the authors show that their VQE kernel is able to outperform other kernels in a BO setting, and that their NFT-EMICoRe approach is able to outperform other (non-BO) NFT approaches.

**Strengths:**

The paper tackles an important problem, the optimization of noisy VQE circuits, and offers a principled solution using BO combined with physical constraints, using state of the art methods (NFT).  The paper is well written, and the experimentation is well chosen to support the method.

**Weaknesses:**

The experimentation could be expanded.  Particularly, it would be interesting to see how the model performs on an actual quantum implementation.  Also, investigation of a broader ranger of Hamiltonians would be desirable (including ones motivated by practical problems).

**Questions:**

The threshold parameter \kappa is introduced in Sec. 3.2, but doesn't appear to be explored in the experimentation.  In particular, did the authors verify that an intermediate value of this parameter is advantageous (hence supporting the use of the EMICoRe acquisition function)?  I may have missed this in the experimentation.

Further, on lines 133-134, they state that they are not concerned with circuit noise on current NISQ devices, but line 320 states that the experimentation confirms the suitability of their method for such devices - is such noise explicitly included in the simulations?

**Limitations:**

Yes, limitations are addressed.

---

> ### Author Rebuttal · Authors · 2023-08-07
>
> We thank the referee for their insightful comments and positive feedback on our work. We take this opportunity to address and respond to the comments below:
>
> ## **Weaknesses**
> > The experimentation could be expanded. Particularly, it would be interesting to see how the model performs on an actual quantum implementation. Also, investigation of a broader range of Hamiltonians would be desirable (including ones motivated by practical problems).
>
> We agree that our method is suited for a broader range of applications as well as Hamiltonians. The Ising and Heisenberg Hamiltonians, however, represent standard benchmarks that are of high practical relevance. They correspond to spin chain Hamiltonians that are widely studied in condensed matter physics, see e.g. Refs. [1,2]. Furthermore, many lattice field theories can be represented as generalized spin chains (see, e.g., Eq. (4)-(6) in [3]). As far as the implementation on actual quantum devices is concerned, the standard benchmarking is usually performed on simulated, noiseless devices which only account for shot noise. We plan to investigate hardware noise in future work. For more details please see the fourth bullet point in the general rebuttal.
>
> ## **Questions**
> > The threshold parameter $\kappa$ is introduced in Sec. 3.2, but doesn't appear to be explored in the experimentation. In particular, did the authors verify that an intermediate value of this parameter is advantageous (hence supporting the use of the EMICoRe acquisition function)? I may have missed this in the experimentation.
>
> Thank you for raising this crucial question. We performed an additional investigation for different heuristic for setting the parameter $\kappa$. This study can be found in the general rebuttal, see Table R1, and will be added to the appendix of the paper in its updated version.
>
> > Further, on lines 133-134, they state that they are not concerned with circuit noise on current NISQ devices, but line 320 states that the experimentation confirms the suitability of their method for such devices - is such noise explicitly included in the simulations?
>
> In our experiments, we only consider shot noise throughout the paper. We apologize for any confusion our phrasing may have caused. We will revise the main text to make this clearer.
>
> ## **References**
>
> - [1] [Funcke L. et al., arXiv preprint arXiv:2302.00467 (2023)](https://arxiv.org/pdf/2302.00467.pdf)
> - [2] [Di Meglio A. et al, arXiv preprint arXiv:2307.03236 (2023).](https://arxiv.org/pdf/2307.03236.pdf)
> - [3] [Atas Y. et al, Nature communications 12, no. 1:6499 (2021).](https://arxiv.org/pdf/2102.08920.pdf)

---

> > ### Comment · Reviewer_WANE · 2023-08-19
> >
> > I thank the authors for their clarifications, and I agree that the new results on tuning $\kappa$ are a valuable addition to the paper.

---

### Official Review · Reviewer_93Tb · 2023-07-19

**Soundness:** 3 good
**Presentation:** 4 excellent
**Contribution:** 2 fair
**Rating:** 6
**Confidence:** 3

**Summary:**

I have reviewed the rebuttal, and I intend to maintain my decision. I appreciate the authors made the efforts to clarify most of my concerns. However, I believe it is essential to address the issue of hardware noise in the present study, a point that previous research seems to have overlooked. I strongly encourage the authors to delve deeper into this matter, as a robust algorithm should be capable of practical implementation on real quantum computers. I believe the issue of hardware noise is actually an inescapable and important challenge here.

This manuscript presents an innovative approach known as EMICoRe, which leverages the synergy between Bayesian optimization and prior knowledge of variational quantum eigensolvers (VQE) to enhance the efficiency of the optimization. The authors introduce a novel kernel specifically designed for Bayesian optimization and employ it within the EMICoRe framework. Additionally, the paper conducts numerical experiments to compare the proposed method with the state-of-the-art baselines, demonstrating its superior performance.

**Strengths:**

- The manuscript introduces a novel optimization algorithm based on Bayesian optimization that incorporates the specific property of the VQE objective function into its design.
- The paper is well-organized and clearly written. The figures in the paper effectively illustrate the improved performance achieved by the proposed method in comparison to the baselines.
- The paper establishes the equivalence between two previously proposed VQE properties, providing a foundational basis for the subsequent analysis on VQE objective functions.

**Weaknesses:**

- The numerical results presented in the paper indicate that the proposed method does not exhibit a clear advantage when the number of observed points is limited, as compared to the baselines.
- Although the proposed method exhibits improved performance compared to the baseline as the number of observed points increases, it still falls short of achieving the ideal ground state.
- The proposed method just considers the shot noise but ignores possible noise originating from the quantum circuit itself.

**Questions:**

1. Based on the numerical results presented in the paper, the fidelity between the ground state and the state obtained through the VQE remains consistently below 0.8, even after reaching 400 observed points. It is noteworthy that the VQE circuit being optimized in the study comprises only three layers and five qubits. Therefore, I believe it is crucial to investigate the potential impact of further increasing the number of observed points and determine the minimum number of observed points required to achieve a higher fidelity, such as surpassing 0.95.

2. The manuscript just includes two parameter settings, one for the Ising model and the other for the Heisenberg model. However, to verify the robustness of the proposed method, I suggest that the authors incorporate more scenarios for validation.

**Limitations:**

I believe the authors have provided thorough discussions on the limitations of their work within the Appendix.

---

> ### Author Rebuttal · Authors · 2023-08-07
>
> We thank the reviewer for the valuable feedback and appreciation of our manuscript.
>
> ## **Weaknesses**
> > The numerical results presented in the paper indicate that the proposed method does not exhibit a clear advantage when the number of observed points is limited, as compared to the baselines.
>
> Indeed, when the number of observed points is small the GP surrogate model is not informative enough to provide an effective acquisition function. When the number of observed points gets larger, and thus the GP surrogate model becomes more informative, our EMICoRe approach starts outperforming standard baselines, i.e., our informed search for new points to observe, based on the CoRe, shows its benefit.
>
> > Although the proposed method exhibits improved performance compared to the baseline as the number of observed points increases, it still falls short of achieving the ideal ground state.
>
> We would like to point out that the primary objective of our work is to demonstrate that EMICoRe can improve the efficiency of the training process, and not to extract the physics of the Ising model, which is well known in the literature. Thus, we compare the performance of the training process for different methods under a fixed budget of observations. Our method reaches the ground state by simply performing longer optimizations. To demonstrate this, we ran additional experiments with a larger number of observations, see Fig. R2 in the PDF. From this plot, it is evident that taking more observations will make our EMICoRe converge to the ground state as expected.
>
> > The proposed method just considers the shot noise but ignores possible noise originating from the quantum circuit itself.
>
> When benchmarking the performance of newly proposed hybrid quantum-classical algorithms, the noise from quantum hardware is usually not included in first-case studies (see e.g., [1,2,3]). A standard procedure in the field is to benchmark first on shot noise only, see the NFT [1] experiments, and consider hardware noise in potential follow-up studies. As stated in the general rebuttal, the main reason is that quantum noise is strongly hardware dependent, e.g., superconducting quantum hardware is affected by fundamentally different types of noise (in particular, CNOT gate noise, decoherence, and measurement noise) compared to, e.g., trapped-ion quantum hardware. Moreover, various error mitigation schemes exist for different types of quantum hardware, which generally require additional error calibration runs on the hardware. An informative study of quantum hardware noise would need to take all these considerations into account and therefore is beyond the scope of the initial proposal of our novel method.
>
> ## **Questions**
> > Based on the numerical results presented in the paper, the fidelity between the ground state and the state obtained through the VQE remains consistently below 0.8, even after reaching 400 observed points. It is noteworthy that the VQE circuit being optimized in the study comprises only three layers and five qubits. Therefore, I believe it is crucial to investigate the potential impact of further increasing the number of observed points and determine the minimum number of observed points required to achieve a higher fidelity, such as surpassing 0.95.
>
> We agree with the referee and we have performed (longer) experiments, see Fig. R2, in the attached PDF. In this analysis, we show that running the optimization longer, i.e., performing more observations, lets the energy converge to the ground state and the fidelity to approach $0.976$.  Looking at the figure, we also note that EMICoRe requires 1000 observations in order to push the average fidelity above $0.95$, while NFT never reaches that value. In Table R2 below, we report the mean and standard deviation for the energy and the fidelity for the EMICoRe and NFT baselines when running for 6000 observations, associated to the experiments mentioned above.
>
> > The manuscript just includes two parameter settings, one for the Ising model and the other for the Heisenberg model. However, to verify the robustness of the proposed method, I suggest that the authors incorporate more scenarios for validation.
>
> Our experiments focus on the Heisenberg model with the Ising model as a special subclass of the Heisenberg one. In both cases, the parameters are chosen such that in the thermodynamic limit, they correspond to critical points, i.e., where a quantum phase transition occurs. This choice of setup is in general among the most challenging scenarios, as the corresponding ground states show the largest amount of entanglement. We thus expect any other choice of parameters to render the problem easier. To complement our results, we added another choice of couplings in the PDF document, see Fig. R1. We plan to further extend our study to other Hamiltonians, e.g., quantum chemistry systems, in future work.
>
>
> ### Table R2: Energy and fidelity for longer runs (6000 observations) for the Ising model at criticality. Best results highlighted in bold. For energy, lower is better. For the fidelity, higher is better.
> | kernel             |  Energy  | Fidelity |
> |--------------------|----------|----------|
> | **EMICoRe (ours)** | **-5.973762 +- 0.045612**   | **0.975552 +- 0.039288** |
> | NFT-random | -5.929573 +- 0.075909    |  0.918256 +- 0.097190|
> | NFT-sequential |  -5.930662 +- 0.097275    |  0.916873 +- 0.155590 |
>
> ## **References**
>
>
> - [1] [Nakanishi K. et al., Phys. Rev. Research 2, 043158 (2020).](https://journals.aps.org/prresearch/abstract/10.1103/PhysRevResearch.2.043158)
> - [2] [Farhi E. et al., arXiv:1411.4028 (2014)](https://arxiv.org/abs/1411.4028)
> - [3] [Bravo-Prieto C. et al., Quantum 4, 272 (2020).](https://quantum-journal.org/papers/q-2020-05-28-272/)

---

### Author Rebuttal · Authors · 2023-08-07

We thank the four reviewers for their valuable feedback. To streamline our reply, we place the referenced tables at the bottom.

- Some reviewers suggested additional experiments for other parameter choices of the target Hamiltonians. We stress that the Ising Hamiltonian, which is studied along with the Heisenberg Hamiltonian, is considered at criticality (in our paper) and thus represents a challenging optimization objective. Studying the Hamiltonians for coupling values off criticality can be expected to be less challenging. To confirm this, we have performed additional experiments for different coupling parameters. These are shown in Fig. R1 in the attached PDF.

- Furthermore, some reviewers suggested to explore further possibilities of heuristics to set the CoRe threshold $\kappa$. As a result, we investigated this as part of additional experiments. Specifically, we used the following updating rule for tuning the value of $\kappa$ at each step:
$$
\kappa = \max\left(C_0\cdot\sigma, C_1\cdot\frac{ \hat{\mu}^{t - T_{\mathrm{Ave}}} - \hat{\mu}^{t}}{T_{\mathrm{Ave}}}\right)
$$with hyperparameters $C_0$ and $C_1$. We note that $\sigma$ is the standard deviation of the observation noise, and fixing $C_0=0, C_1=1.0$ reduces to the heuristics used in the original submission.
We report the results for different choices of the hyperparameters $C_0$ and $C_1$ in Table R1 below. The best combination is highlighted in bold. The experiments were performed for the Ising model at criticality using EMICoRe and a $(L = 3)$-layered $(Q = 5)$-qubits quantum circuit with $N_\textrm{{shots}}=1024$. Each trial ran for 600 observations. Table R1 demonstrates that intermediate values for the hyperparameters $C_0, C_1$ achieve the best result, thus proving the effectiveness of EMICoRe. Also, we found that the combination $C_0=0.1, C_1=10$ improves the result from the original submission. We thank the reviewers for the suggestion, and we will update the results in the paper with the new heuristics.

- Some reviewers also asked why the ground state is never reached by our method as well as any baseline. This is merely a consequence of taking a limited number of observations, and we performed longer optimizations to confirm this. Specifically, Fig. R2 in the attached PDF demonstrates that running the experiments longer, i.e., observing more points, makes the optimization converge closer to the ground state and raises the fidelity to  $0.976$, see table R2 in the reply to referee 93Tb for details. We explicitly tested that the same holds true for every other choice of parameters and circuit setups investigated in this manuscript, e.g., $(Q,L)=(3,3), (5,3), (7,5)$.  For each of those setups, we achieved similarly high fidelities. We will add these experiments to the revised manuscript.

- Another question raised by some referees relates to the type of noise investigated in our work. When benchmarking the performance of newly proposed hybrid quantum-classical algorithms, the noise from quantum hardware is usually not included in first-case studies. A standard procedure in the field is to benchmark first on shot noise only and consider hardware noise in potential follow-up studies, see, for example, the NFT paper [1] and the other Refs. [2,3] below. The main reason is that quantum noise is strongly hardware dependent, e.g., superconducting quantum hardware is affected by fundamentally different types of noise (in particular CNOT gate noise, decoherence, and measurement noise) compared to, e.g., trapped-ion quantum hardware. Moreover, various error mitigation schemes exist for different types of quantum hardware, which generally require additional error calibration runs on the hardware. An informative study of quantum hardware noise would need to take all these considerations into account and therefore is beyond the scope of the initial proposal of our novel method.

Due to some potential misunderstandings by a reviewer, we would like to kindly reiterate the main contributions of our work as follows:

- We propose a novel classical kernel, the VQE kernel, which is uniquely suited for the VQE setup. Specifically, its corresponding feature vectors are the basis of the energy functions that can be modeled by the parameterized quantum circuit.
- Leveraging this powerful inductive bias allows us to propose a novel Bayesian optimization method, EMICoRe, which harnesses the fact that for the parameterized quantum circuit, only three observations allow us to determine the energy landscape along an entire line.
- We demonstrate in our numerical experiments that our proposed method can outperform the current state-of-the-art method NFT and standard BO schemes.
----------
### Table R1: Results for a new heuristic of setting $\kappa$ using the hyperparameters $C_0$ and $C_1$. Best results highlighted in bold. For energy, lower is better. For fidelity, higher is better.
| Description     | $C_0$  |   $C_1$     | Energy | Fidelity |
|-----------------|-------------------|---------------------|--------|----------|
| In Manuscript   | 0.0               | 1.0    | -5.817205 ± 0.140868  | 0.857839 ± 0.159668 |
| Extreme (small) | 0.1               | 0.1     | -5.816213 ± 0.113959  | 0.857070 ± 0.163497 |
| High (large)    | 10.0              | 10.0    | -5.719012 ± 0.152881  | 0.824516 ± 0.156895 |
| Extreme (large) | 10.0             | 100.0     | -5.703273 ± 0.156716 | 0.804686 ± 0.179101 |
| **Best**   | **0.1**          | **10.0**   | **-5.842853 ± 0.089015**   | **0.869609 ± 0.112713**|

## References

- [1] [Nakanishi K. et al., Phys. Rev. Research 2, 043158 (2020).](https://journals.aps.org/prresearch/abstract/10.1103/PhysRevResearch.2.043158)
- [2] [E. Farhi et al., arXiv:1411.4028 (2014)](https://arxiv.org/abs/1411.4028)
- [3] [Carlos Bravo-Prieto et al., Quantum 4, 272 (2020).](https://quantum-journal.org/papers/q-2020-05-28-272/)

---

### Decision · Program_Chairs · 2023-09-21

**Decision:**

Accept (poster)

**Comment:**

The authors proposed a novel method, known as EMICoRe, to combine Bayesian optimization and prior knowledge of variational quantum eigensolvers (VQE) to enhance the efficiency of the optimization. During the rebuttal, the authors have addressed all the concerns raised by the reviewers. The authors have also provided additional experiments to demonstrate the performance of the proposed method. All the reviewers appreciate the contributions and vote favorably for acceptance. Therefore, I recommend the acceptance of the paper.